# Molecular interactions between Hel2 and RNA supporting ribosome-associated quality control

Marie-Luise Winz[1], Lauri Peil[1,3], Tomasz W. Turowski[1], Juri Rappsilber [1,2] & David Tollervey [1]

Ribosome-associated quality control (RQC) pathways monitor and respond to ribosome stalling. Using in vivo UV-crosslinking and mass spectrometry, we identified a C-terminal region in Hel2/Rqt1 as an RNA binding domain. Complementary crosslinking and sequencing data for Hel2 revealed binding to 18S rRNA and translated mRNAs. Hel2 preferentially bound mRNAs upstream and downstream of the stop codon. C-terminal truncation of Hel2 abolished the major 18S crosslink and polysome association, and altered mRNA binding. *HEL2* deletion caused loss of RQC and, we report here, no-go decay (NGD), with comparable effects for Hel2 truncation including the RNA-binding site. Asc1 acts upstream of Hel2 in RQC and *asc1Δ* impaired Hel2 binding to 18S and mRNA. In conclusion: Hel2 is recruited or stabilized on translating 40S ribosomal subunits by interactions with 18S rRNA and Asc1. This 18S interaction is required for Hel2 function in RQC and NGD. Hel2 probably interacts with mRNA during translation termination.

[1] Wellcome Centre for Cell Biology, University of Edinburgh, Michael Swann Building, Kings Buildings, Mayfield Road, Edinburgh EH9 3BF, Scotland. [2] Bioanalytics, Institute of Biotechnology, Technische Universität Berlin, 13355 Berlin, Germany. [3]Present address: Institute of Technology, University of Tartu, Nooruse 150411 Tartu, Estonia. Correspondence and requests for materials should be addressed to D.T. (email: d.tollervey@ed.ac.uk)

Eukaryotes respond to stalled ribosomes via two major, conserved surveillance pathways: non-stop decay (NSD) and no-go decay (NGD) (reviewed in refs. [1,2]), which ultimately degrade the mRNA. NSD is triggered by ribosomes stalled at the very 3' end of an mRNA in the absence of translation termination, typically due to aberrant pre-mRNA cleavage or stop codon mutation[3,4]. NGD[5] is typically caused by ribosome stalling within the open reading frame (ORF), resulting from mRNA secondary structure, RNA damage, aminoacyl-tRNA deficiency or translation through polybasic stretches[6,7]. Stretches of specific rare codons, such as CGA[7,8] (coding for arginine) or AAA[6,9–12] (lysine, encountered during translation of the poly(A) tail), also trigger ribosome stalling. Ribosome stalling risks producing truncated, potentially toxic, products and depletes functional ribosomes. Moreover, re-initiation of translation following stalling can lead to frameshifting[8,10,11]. The rapid identification and dissociation of stalled complexes, and degradation of stalling-prone mRNAs, are therefore important activities. Ribosome stalling also triggers another surveillance pathway, termed ribosome-associated quality control (RQC; reviewed in refs. [2,13]), which targets truncated nascent peptides to proteasomal degradation via ubiquitination[14,15]. Each of these pathways encompasses detection of stalling, splitting of 80S ribosomes[16–18] and degradation of either faulty mRNA or nascent peptide, whereas the ribosomal subunits and any associated tRNAs are recycled. However, it remains unclear exactly how translational aberrations are sensed and downstream responses triggered.

Recent research has focussed on two early-acting factors: Asc1 (RACK1 in mammals) is required for NGD[19], NSD[20] and RQC[21] and is an integral component of the ribosomal 40S subunit. Hel2 (also known as Rqt1 in yeast; ZNF598 in mammals) is a RING-type E3 ubiquitin ligase that is required for RQC[21–23]. Moreover, while Hel2 was reported to be ribosome associated[24], it has only ~0.6% of the abundance of ribosomes[25,26]. Both Hel2 and human ZNF598 (hZNF598) are non-essential but their deletion increases full-length (FL) translation on stalling-prone, poly(CGA)- or poly(A)-containing mRNAs[7,21] and modestly reduces ribosome stalling on reporter constructs[23]. A group of Hel2-associated proteins, Slh1/Rqt2, Cue3/Rqt3 and Ykr023W/Rqt4, also function in triggering RQC, but their precise roles remain unclear[22,23].

Following the triggering of surveillance, the E3 ubiquitin ligase Ltn1 modifies the nascent polypeptide with K48-linked poly-ubiquitin[14], targeting it for proteasome-mediated degradation. In contrast, hZNF598 ubiquitinates the 40S ribosomal subunit, with major substrates at two lysine residues in Rps10 (eS10) and additional sites in Rps3 (eS3) and Rps20 (uS10)[10–12], while Hel2 in *Saccharomyces cerevisiae* ubiquitinates Rps20 (uS10) and Rps3 (eS3)[22]. K48-linked, single and double ubiquitination have been observed for Rps3[22]. Previous work also implicated Hel2 in K63-linked polyubiquitination, important for turnover of truncated peptides following translation stalling[27]. An additional activity for Hel2 is the clearance of ribosomes that have undergone incomplete maturation[28]

Hel2 mutants are sensitive to histone overexpression, giving rise to the designation "Histone E3 ubiquitin Ligase 2"[29]. However, our data for Hel2 crosslinking to RNA offer no support for a nuclear function of Hel2. The major targets of Hel2 binding are within 18S ribosomal RNA, and this interaction is lost upon deletion of the crosslinked region from Hel2, which also leads to a loss of polysomal association, RQC and NGD. Further interactions with mRNAs upstream and downstream from the stop codon, as well as with tRNAs, placed Hel2 on translating and terminating ribosomes.

## Results

**The unstructured C-terminus of Hel2 contacts RNA.** Inspection of the sequence of Hel2 did not reveal any evident RNA-binding motif or interaction domain. Regions of RNA-binding proteins that make direct contact with RNA can be identified by ultraviolet (UV) crosslinking followed by mass spectrometry (MS), to detect and characterize the covalent nucleotide–peptide conjugate[30]. In the recently reported identification of RNA-associated peptides (iRAP) technique[31], RNA–protein crosslinking sites can be identified with amino acid resolution from all RNA classes. Briefly, RNA–protein complexes are covalently crosslinked with 254 nm UV irradiation in actively growing yeast cells (Fig. 1a). Following initial protein digestion with LysC, RNA–protein

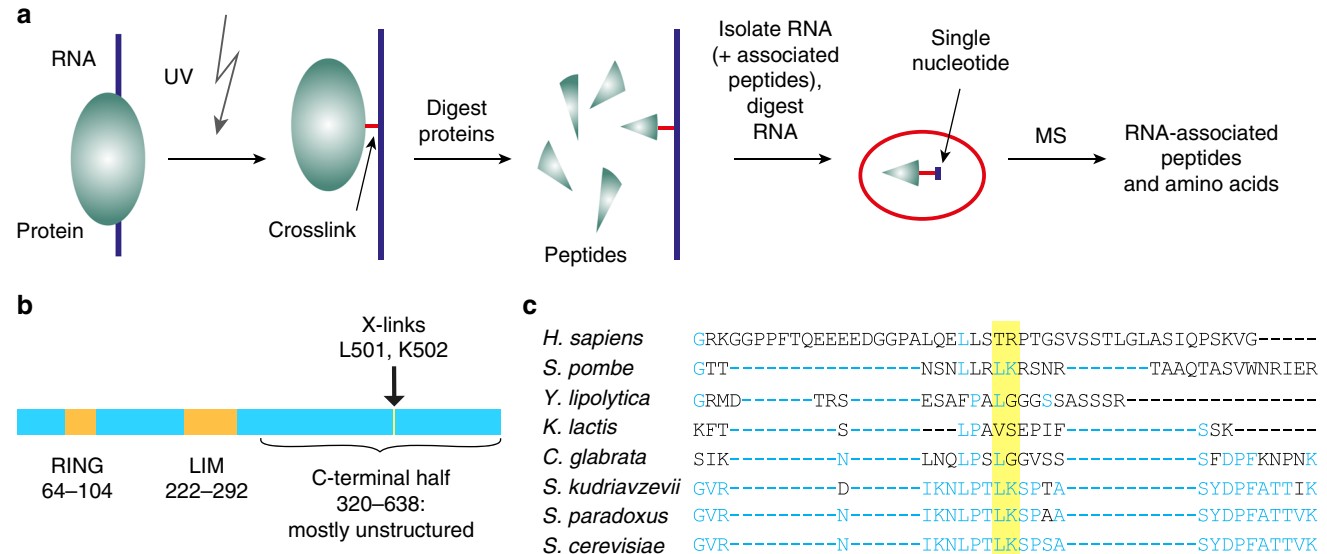

**Fig. 1** iRAP and iRAP-identified Hel2 crosslink sites, sequence and structure elements and conservation. **a** Schematic representation of the iRAP procedure. **b** Positioning of sequence and structural elements as well as crosslinked amino acids revealed by iRAP within the Hel2 primary structure. See Supplementary Fig. 3 for a structure prediction of Hel2. **c** Multiple sequence alignment of crosslink-proximal amino acids of Hel2 from *S. cerevisiae*, *S. paradoxus*, *S. kudriavzevii*, *C. glabrata*, *K. lactis*, *Y. lipolytica* and *S. pombe* with *H. sapiens* ZNF598. The two crosslinked amino acids and corresponding amino acids in homologues are highlighted in yellow. Conserved amino acids are shown in blue (conservation w.r.t. *S. cerevisiae*)

conjugates are separated from proteins by centrifugation through a density gradient of caesium trifluoroacetate. Following RNA digestion, peptides are analysed by liquid chromatography (LC)-MS/MS analysis, searching for the specific addition of mass corresponding to the residual mononucleotide that remains following nuclease digestion[31]. These analyses identified a single peptide from Hel2 in the RNA-associated fraction, which carried additional mass corresponding to crosslinked uracil. Two adjacent amino acids were identified as crosslinked, $L_{501}$ and $K_{502}$ (Fig. 1b), located in the C-terminal region of Hel2 (full length: 639 amino acids). The RNA-binding site is distant from the RING finger ubiquitin ligase domain (64–104) and the LIM domain, a putative protein interaction site (222–292) (Fig. 1b), which are highly conserved between human and yeast (Supplementary Fig. 1).

The amino acid sequence in the C-terminal domain (CTD) of Hel2, including the region around the crosslinked site, is not highly conserved between Hel2 and hZNF598, possibly related to altered ribosomal protein substrate specificity (Fig. 1c and Supplementary Fig. 2). Despite low sequence conservation, CTD regions of both the yeast and human[12] proteins are predicted to be highly unstructured (Supplementary Fig. 3) and to bind RNA, since complete deletion of the CTD blocked RNA binding by hZNF598[12].

**Hel2 has major binding sites in 18S rRNA and mRNA.** To identify RNAs bound by Hel2 in vivo, we applied crosslinking and analysis of cDNAs (CRAC)[32]. For this, a C-terminal fusion between Hel2 and His$_6$-TEV-ProteinA$_2$ (HTP) (Fig. 2a) was expressed from the HEL2 locus under the control of the

endogenous $P_{HEL2}$ promoter. The tagged protein is the only form of Hel2 in the cell.

Hel2 is non-essential, so the functionality of Hel2-HTP was confirmed by growth tests in the presence of the translation inhibitors azetidine-2-carboxylic acid (AZC) and anisomycin, to which hel2Δ strains are hypersensitive[33,34] (Supplementary Fig. 4). We also tested hydroxyurea (HU) resistance, which is decreased by impaired nonsense suppression[35] and loss of Hel2[36,37] or Asc1[36–39]. Compared to two independent hel2Δ strains, these data indicated that HTP-tagged versions of Hel2 retain substantial functionality and are likely to form interactions similar to wild-type Hel2 (Supplementary Fig. 4).

CRAC was initially performed on six Hel2-HTP strains derived from independent clones and four untagged controls, in a total of four independent experiments (see Supplementary Tables 1-3 for more information and aligned read numbers). Analysis of the sequence data (Fig. 2b, Supplementary Table 4) identified the major RNA target classes for Hel2 binding as mature rRNA and mRNA, with lower tRNA association, consistent with previously reported polysome association of Hel2[22]. tRNA recovery with Hel2-HTP was reproducibly higher than in the non-tagged control but without clear enrichment for specific tRNA species. In contrast, PAR-CLIP analysis of hZNF598 reported enrichment of specific tRNAs[12].

Mapping reads to RDN37, which encodes the polycistronic pre-rRNA transcript, showed strong binding within the 18S rRNA (Fig. 3a), with one major and two minor peaks. The major and one minor binding site are located in the 3' domain of 18S rRNA, with the remaining minor site in the 5' domain. Crosslinks were also identified over the 18S 5' domain in untagged strains but with substantially lower frequency (Fig. 3a).

The precise crosslinked nucleotides can be identified by the locations of mutations in the cDNA sequences, particularly single-nucleotide micro-deletions that are not commonly generated by reverse transcription (RT), polymerase chain reaction (PCR) or sequencing errors[32]. This localized the crosslinks in all three 18S peaks to single-stranded nucleotides within stem-loop structures (C1490-A1492 for the major, and U494 and U1362 for the 5' and 3' minor peaks, respectively; Fig. 3b)[40]. In addition, we observed a minor crosslink site within the 5.8S rRNA, which forms part of the exit of the peptide exit tunnel. This might not only indicate Hel2 interactions with nascent peptides exiting the ribosome but could also reflect interactions between nascent Hel2 and the ribosome during translation.

Localization of the 18S crosslinking sites within a 3D structure of the S. cerevisiae ribosome (Fig. 3c)[41] indicates that Hel2 is positioned on the head domain of the small subunit, almost centrally between the mRNA entry and exit channels. The crosslinks in the 3' domain are closer to the entry channel, whereas the minor crosslink in the 5' domain is positioned closer to Asc1, which is localized near the mRNA exit channel. The position of the major crosslink site also suggested that Hel2 can contact ribosomal proteins around the mRNA entry channel. Notably, a major Hel2-crosslinked nucleotide (C1490) is directly contacted by the N-terminus of Rps3, with Rps10 and Rps20 in close proximity (14.5 Å and 19.7 Å)[41].

In S. cerevisiae, Rps20 is the major Hel2 ubiquitination target and the major Hel2–rRNA crosslink is located very close to the target lysines (Fig. 3c). In Rps3, K212 is a minor target for Hel2[22] and corresponds to K214 in hRPS3, ubiquitinated by hZNF598[10–12]. However, the major targets for ubiquitination by hZNF598 lie in the C-terminus of hRPS10 (K138/139)[10–12], which is absent in S. cerevisiae, while the minor modified K107 in hRPS10 is replaced by arginine (Supplementary Fig. 5).

We speculate that the loss of the Rps10 C-terminus in budding yeasts drove accompanying changes in Hel2/ZNF598, causing

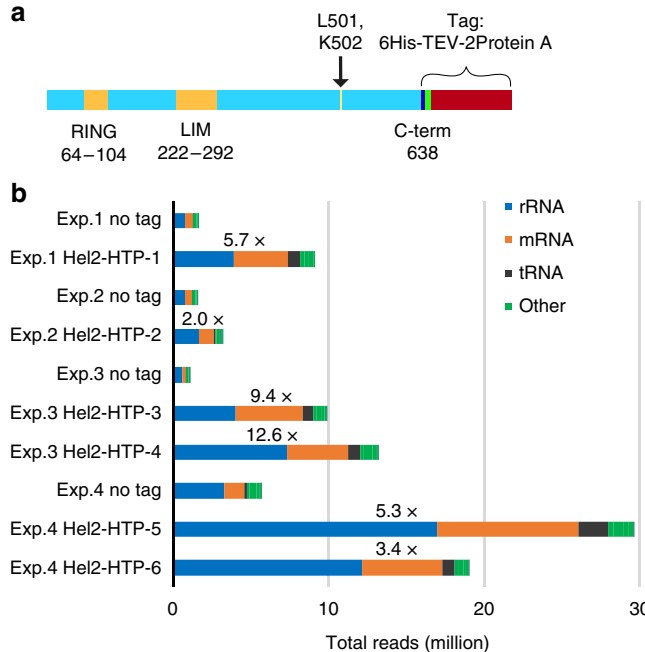

**Fig. 2** Hel2 construct for crosslinking and analysis of cDNA (CRAC) and read distribution. **a** C-terminally His$_6$-TEV-ProteinA$_2$ (HTP)-tagged Hel2. RING and LIM domains and crosslinked amino acids are highlighted. Representation approximately in scale. **b** Read distributions compared between four replicates of CRAC with (wild-type) Hel2-HTP and untagged controls, showing total reads per transcript class in each data set, with fold-enrichment (of total RNA reads) in Hel2-HTP CRAC compared to untagged control indicated above corresponding bars. See also Supplementary Table 4

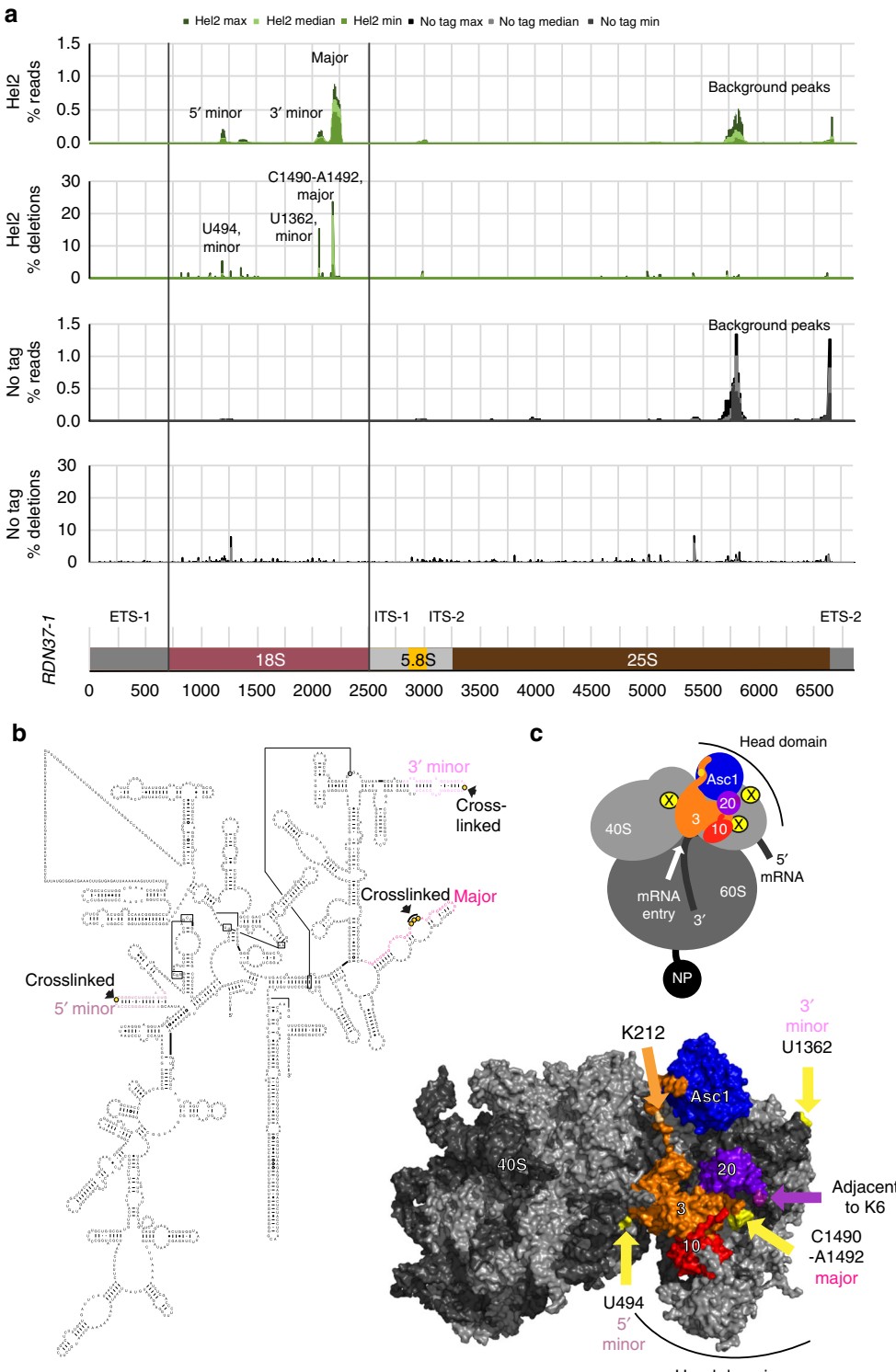

**Fig. 3** Hel2 ribosomal crosslinking sites. **a** Hel2 and untagged crosslinking and analysis of cDNA (CRAC) data aligned to *RDN37-1* showing reads (first and third panel) and deletions (second and fourth panel) all normalized to total reads/deletions in *RDN37-1* (fifth panel). ETS external transcribed spacer, ITS internal transcribed spacer. **b** 18S rRNA secondary structure according to Comparative RNA Website (CRW)[40] with stem-loop structures enriched in Hel2 CRAC highlighted in different shades of pink and crosslinked nucleotides (high-frequency micro-deletions in Hel2 CRAC) highlighted in yellow. **c** Crosslink sites (high frequency deletions in Hel2 CRAC) highlighted on a 3D structure representation of the *S. cerevisiae* ribosome. Top panel: schematic model showing the locations of the nascent peptide position and crosslinks (major: **X**, minor: x). Lower panel: crystal structure[41]; pdb 4v88, 40S subunit of monosome chain B is shown) with crosslinked nucleotides highlighted in yellow and further indicated by yellow arrows. In both panels, specific ribosomal proteins are indicated: Asc1 (dark blue), Rps3 (orange), Rps10 (red), Rps20 (violet). Other ribosomal proteins are shown in lighter grey, with rRNA in darker grey. Ubiquitination target sites are indicated by coloured arrows; violet for Rps20 (K6) (adjacent site shown, since the actual amino acid is not part of the crystal structure) and orange for Rps3 (K212)

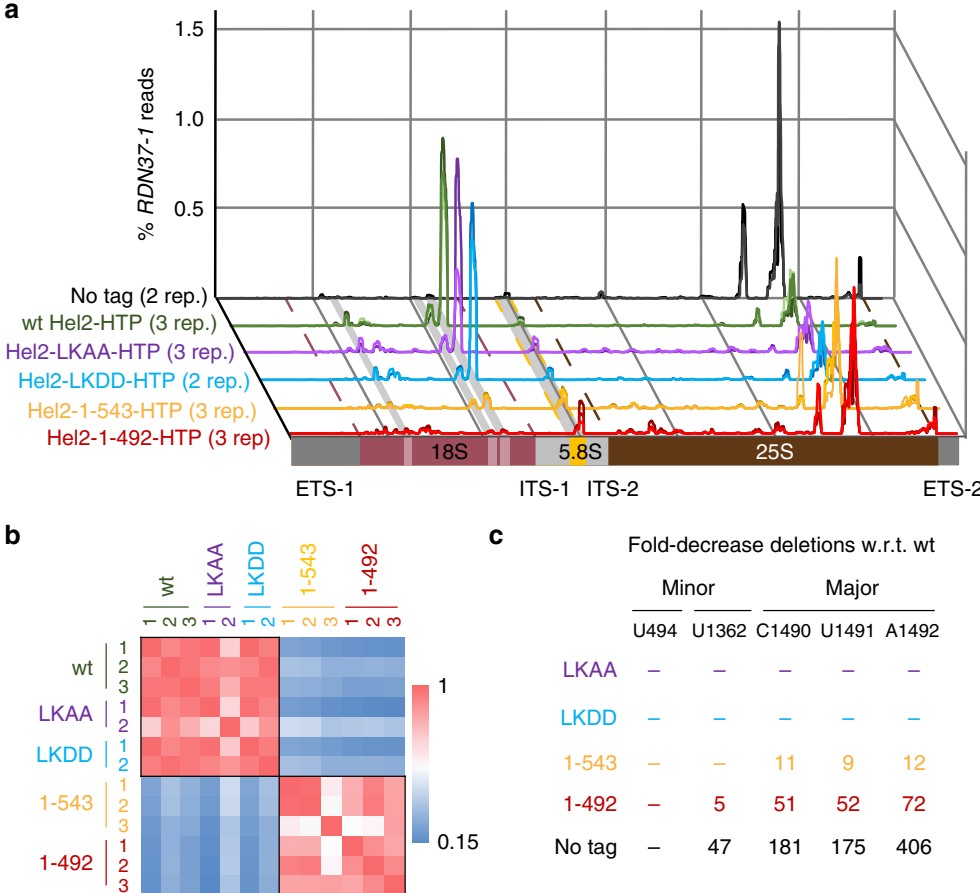

**Fig. 4** Effects of Hel2 mutations on binding of ribosomal RNA. **a** Relative distribution of reads over *RDN37-1* gene encoding three of the four ribosomal RNAs. Major and minor 18S peaks and the 5.8S peak are highlighted with grey bars. The 25S peak has been seen in many different crosslinking and analysis of cDNA (CRAC) analyses and appears to be non-specific. Borders of 18S and 25S rRNAs are indicated as dashed lines. The sum of all reads over *RDN37-1* is normalized to 1 for each sample. See Supplementary Fig. 6 for schematic depiction of mutants. **b** Pearson correlation between binding patterns (reads) in different samples. **c** Change in relative fraction of deletions at previously identified crosslink peaks in 18S rRNA. Minor (U494, U1362) and major (C1490-A1492) peaks are represented. The sum of all deletions over *RDN37-1* was set to 1 for each sample. Medians are only shown for samples of $p < 0.05$ (two-tailed Student's $t$ test, w.r.t. wild-type). See also Supplementary Fig. 7. For U494, a reduced read density was visible in **a**, but the frequency of deletions was not significantly altered, owing to experimental variability in deletion density for the wild-type (Supplementary Fig. 7)

altered ribosome binding and enabling preferential targeting of Rps20. Notably, Rps10 from *Schizosaccharomyces pombe* contains lysine residues that potentially correspond to the major targets in hRPS10, and its Hel2 orthologue more closely resembles the human protein. Conversely, in yeast species that lack the Rps10 ubiquitination targets, Hel2 shows higher homology to *S. cerevisiae* (Supplementary Figs. 2 and 5). We conclude that Hel2 binds to the small ribosomal subunit in close proximity to known ribosomal protein ubiquitination sites.

**Loss of RNA crosslinking to Hel2 disrupts 18S association**. To assess the contribution of the crosslinked region of Hel2 to RNA binding, we initially generated two C-terminal truncations: Hel2$_{1-543}$, which retained the RNA crosslinking sites, and Hel2$_{1-492}$, which lacked these (Supplementary Fig. 6a). Western blot analyses showed that the truncated proteins were more abundant than wild-type Hel2 (~1.8-fold for Hel2$_{1-543}$ and ~1.9-fold for Hel2$_{1-492}$). Both truncated proteins also purified more efficiently in the CRAC procedure (~1.5–2.1-fold more efficiently for Hel2$_{1-543}$ and ~2.4–8.4-fold more efficiently for Hel2$_{1-492}$) as determined following tobacco etch virus (TEV) elution (Supplementary Fig. 6b). Despite this, total recovery of RNA in

association with the mutant proteins was not increased in comparison to wild-type Hel2 (Supplementary Tables 5–7). This indicates reduced RNA-binding efficiency, particularly for Hel2$_{1-492}$.

In CRAC analyses, both truncation mutants showed substantially altered rRNA recovery, with reduced binding at the major site in 18S rRNA (Fig. 4a, b). Focussing on micro-deletions (as a proxy for crosslinking), the loss of the major crosslinks (C1490-A1492) was much more substantial for Hel2$_{1-492}$, which lacked the RNA-crosslinked amino acids, than for Hel2$_{1-543}$ that retained them (Fig. 4c, Supplementary Fig. 7). Additionally, micro-deletions at the minor crosslink site U1362 were five-fold reduced for Hel2$_{1-492}$ but not significantly altered for Hel2$_{1-543}$ (Fig. 4c, Supplementary Fig. 7). For the second minor peak at U494, a reduction in read density was visible (Fig. 4a), but the reduction of micro-deletions was not statistically significant, due to high variability in the wild-type data (Fig. 4c, Supplementary Fig. 7). Relative recovery of the 5.8S peak was increased in Hel2$_{1-492}$ (Fig. 4a), probably reflecting normalization to total mapped reads, after loss of the major binding site.

In tests for altered sensitivity to HU, AZC and anisomycin, Hel2$_{1-492}$ strains showed similar behaviour to the *hel2Δ* (Supplementary Fig. 8). In contrast, Hel2$_{1-543}$, which retained

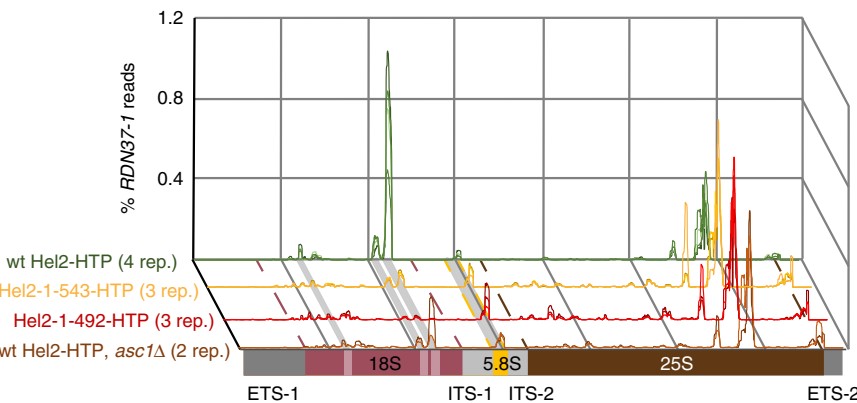

**Fig. 5** Effect of *ASC1* deletion on Hel2 18S binding and comparison to truncation mutants. As Fig. 4a

the RNA interaction domain, showed slightly increased sensitivity to HU but not to AZC or anisomycin. These findings are consistent with the model that Hel2 function requires the interaction of the crosslinked site with 18S rRNA.

We also generated two mutants with the crosslinked amino acids (L501, K502) mutated to alanine (Hel2$_{LKAA}$) or aspartic acid (Hel2$_{LKDD}$). However, 18S crosslinking was broadly unaltered for Hel2$_{LKDD}$-HTP or Hel2$_{LKAA}$-HTP (Fig. 4, Supplementary Fig. 7), indicating the involvement of further amino acids in the RNA interaction. In drug sensitivity screens, Hel2$_{LKAA}$ showed resistance relative to wild-type Hel2 strains, whereas Hel2$_{LKDD}$ did not clearly differ from the wild-type (Supplementary Fig. 8).

**Asc1 is required for interaction of Hel2 with 18S rRNA.** Yeast Asc1 (RACK1 in mammals) has been reported to act with Hel2 in the early events of RQC. Deletion of Asc1 or hRACK1 inhibits the response to translation stalling and ribosomal protein ubiquitination to a similar extent as deletion of Hel2 or hZNF598[11,23]. CRAC was therefore performed in strains expressing Hel2-HTP in an *asc1Δ* background (Hel2-HTP, *asc1Δ*). Here total 18S rRNA binding by Hel2 was substantially reduced (Fig. 5), with microdeletions at the major crosslinking site decreased ~2.5–9-fold (Supplementary Fig. 7). We conclude that Asc1 functions upstream of Hel2 and is required for its normal recruitment to, or retention on, the 18S rRNA.

**18S crosslinks correlate with polysome association.** To test whether altered 18S rRNA crosslinking correlates with changed ribosome association, polysome analyses were performed (Fig. 6). For each sucrose gradient (Fig. 6a), peak fractions were pooled, precipitated and analysed by western blotting to determine the distribution of Hel2, using the glycolytic enzyme Pgk1 as control (Fig. 6b, c). Wild-type Hel2 and Hel2$_{LKAA}$ were predominantly recovered in the polysomal fractions, with mild enrichment on higher-order polysomes. In contrast, Hel2$_{1-492}$ was largely absent from polysomes (but seen on 40S–80S), with the largest amount of protein in the free fraction. The smaller truncation, Hel2$_{1-543}$, showed only a modest reduction (of ~30% relative to wild-type) in polysome association with the greatest difference seen for higher-order polysomes (Fig. 6b, c). In *asc1Δ* strains, the distribution of wild-type Hel2 was broadly similar to the distribution of Hel2$_{1-543}$. This suggests that effects of the smaller truncation in Hel2$_{1-543}$ might reflect impaired interactions with Asc1, whereas the additional defects in the larger truncation Hel2$_{1-492}$ reflect the loss of 18S rRNA binding. In addition, *asc1Δ* strains showed a small but significant loss of higher-order polysomes and increased free ribosomal subunits (Fig. 6a, Supplementary Fig. 9).

**Functional consequences of Hel2 mutations on RQC and NGD.** Strains expressing Hel2 mutations were tested for their ability to carry out RQC and NGD, using a published[23] reporter construct consisting of GFP, a translation stalling or non-stalling sequence, and red fluorescent protein (RFP) (Fig. 7a). In the RQC pathway, the stalled nascent peptide would normally be ubiquitinated by Ltn1 and degraded by the proteasome. In the absence of Ltn1 activity, the stalled nascent peptide can be modified by addition of a carboxy-terminal, alanine- and threonine-containing extension (CAT-tail)[15].

RQC activity was assessed by western blotting of the protein products in strains expressing wild-type or mutant Hel2, with or without Ltn1 (Fig. 7b, quantified in Supplementary Fig. 10). In the *ltn1Δ* background (Fig. 7b, lanes 1–5), in which stalled products do not get ubiquitinated and degraded, the wild-type and Hel2$_{LKAA}$ strains (Fig. 7b, lanes 1, 5) accumulated the expected stalling product and the extended, CAT-tailed product. In the *hel2Δ* strain, the stalling product and the CAT-tails were largely absent (Fig. 7b, lane 2) as previously reported[22,23]. A number of products generated by read-through of the stall site were also visible, including a low level of the FL protein (see also Supplementary Fig. 10). The most abundant species would be consistent with the short read-through products seen in published RiboSeq data[23]. The strain expressing the larger truncation, Hel2$_{1-492}$, closely resembled *hel2Δ* (Fig. 7b, lane 3). With the smaller truncation, Hel2$_{1-543}$, a low level of the apparent CAT-tailed product was consistently detected (Fig. 7b, lane 4 and Supplementary Fig. 10).

In the presence of Ltn1 (Fig. 7b, lanes 6–10, Supplementary Fig. 10), the major product in all strains was the short read-through peptide. In the *hel2Δ* strain, the pattern was similar to the *ltn1Δ* background, with accumulation of read-through and FL peptides. The Hel2$_{1-492}$ strain showed greater accumulation of the FL protein than Hel2$_{1-543}$.

In conclusion, the reporter assay shows that loss of Hel2 domains required for 18S rRNA crosslinking correlated with a marked decrease in RQC. The Hel2$_{1-492}$ truncation, which abolished crosslinking at the major 18S rRNA site, also caused near-complete loss of RQC. The Hel2$_{1-543}$ mutation, with greater residual 18S rRNA crosslinking, showed lower levels of the FL read-through product and detectable CAT-tailed products, indicating greater residual RQC.

The same reporter construct was used to assess NGD by analyses of the RNA products in wild-type and mutant strains (Fig. 7c, Supplementary Fig. 11). The translation stall induces mRNA cleavage by NGD, which is blocked in the absence of Asc1[19]. The 5' cleavage fragment is degraded by the cytoplasmic exosome, aided by helicase Ski2. We therefore deleted *SKI2*, to

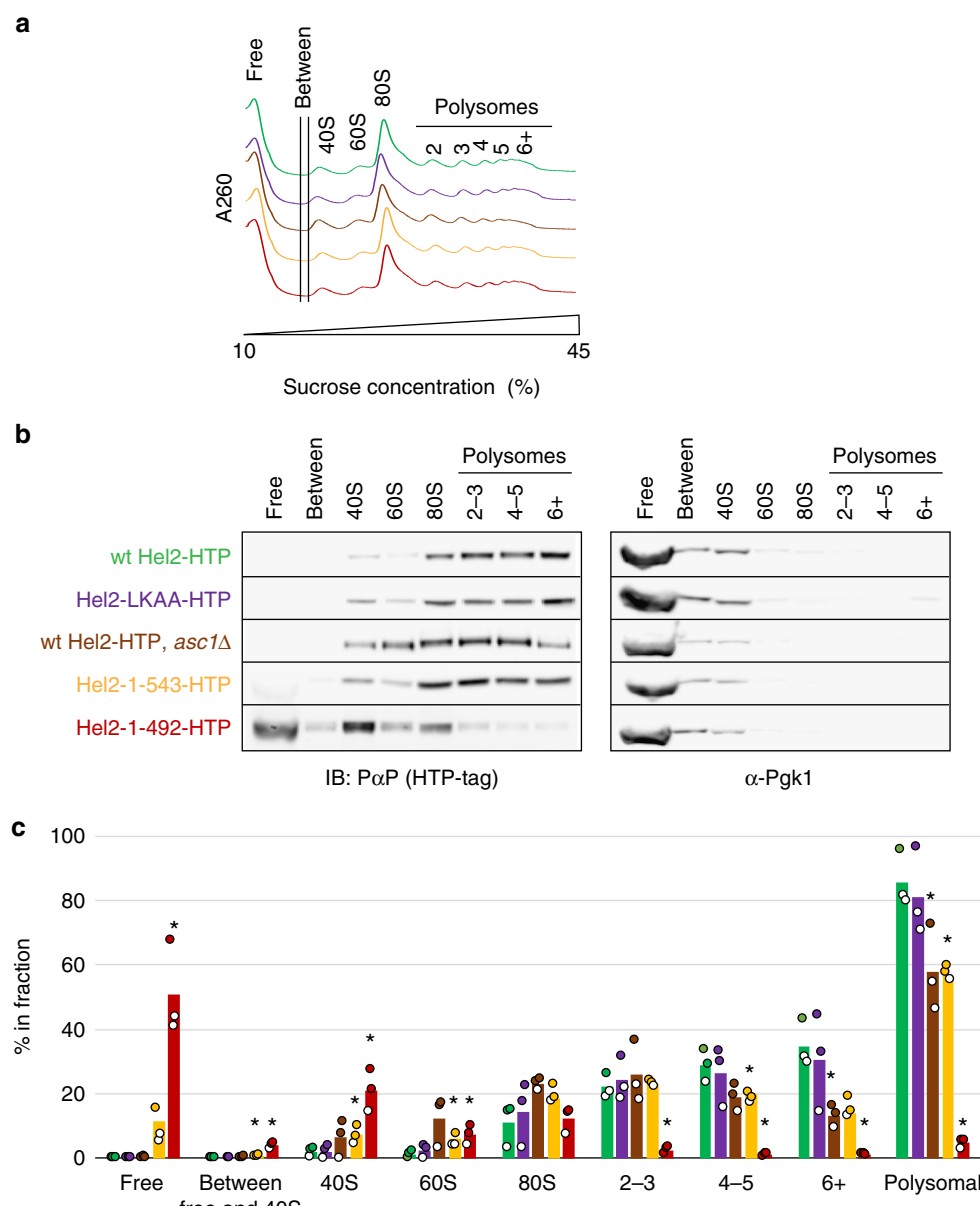

**Fig. 6** Assessment of ribosome and polysome association of different Hel2 mutants. **a** Representative $A_{260}$ profiles (each curve normalized to a sum of 1) of sucrose density gradients from wild-type and mutant strains (see **b** for colour code and Supplementary Fig. 9 for triplicate quantification of signal in different fractions. **b** Immunoblot analysis of tagged Hel2 content in pooled and precipitated sucrose fractions containing different sub-populations of ribosomal components, monosomes or polysomes. Membranes were probed for the HTP-tag (using peroxidase-anti-peroxidase) and for Pgk1 as a control. **c** Quantification of tagged Hel2 content in each pooled fraction. Data were obtained from three independent biological replicates per strain (independently isolated clones). *$p < 0.05$, according to Student's $t$ test (two-tailed, paired, w.r.t. wild-type). See **b** for colour code

allow detection of the cleavage product. Northern analysis was performed using probe Np_GFP that hybridizes at position 352–380 within the GFP ORF, 5' to the stall site.

In wild-type and Hel2$_{LKAA}$ strains, the stalling reporter gave rise to the expected 5'-fragment (Fig. 7c, lanes 1, 5), which was absent from the non-stalling reporter in all strains (Fig. 7c, lanes 6–10). In *hel2Δ* and Hel2$_{1-492}$ strains expressing the stalling reporter, the FL band was stronger than in the wild-type when normalized to the loading control (Supplementary Fig. 11a), indicating less efficient NGD. In addition, heterogeneous bands were visible below the FL transcript and the position of the expected 5'-fragment (Fig. 7c, Supplementary Fig. 11c). The size

distribution of these bands shows a distinct cut-off, consistent with 5' truncation, most likely by Xrn1; the band is no longer detected when the nuclease reaches the probe hybridization site. This activity might serve as a failsafe degradation pathway. The shorter truncation, Hel2$_{1-543}$, showed an intermediate phenotype (Fig. 7c, Supplementary Fig. 11), with reduced but detectable levels of the 5'-fragment seen in the wild-type, together with low but detectable accumulation of the truncated fragments seen in *hel2Δ*.

We conclude that NGD is lost in strains carrying *hel2Δ* or Hel2$_{1-492}$, whereas the Hel2$_{1-543}$ strain retains residual NGD activity.

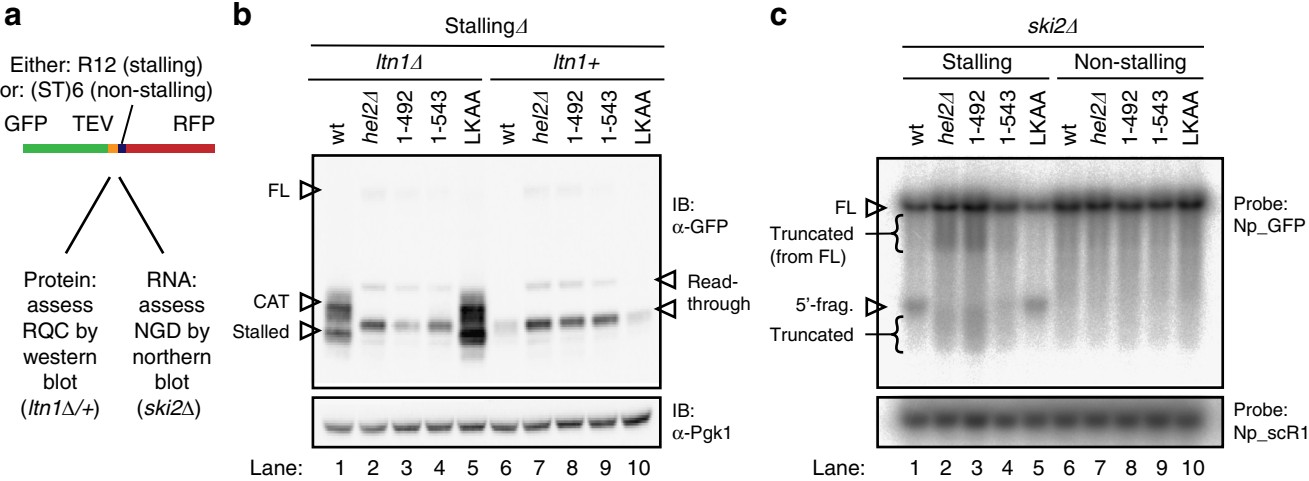

**Fig. 7** Assessing functional consequences of *HEL2* deletion and mutation using a reporter system for ribosome-associated quality control (RQC) and no-go decay (NGD). **a** A published[23] reporter system encoding green fluorescent protein (GFP), TEV protease site, followed by a stalling (R12) or non-stalling ((ST)6) sequence and by red fluorescent protein to assess RQC and NGD. Scheme adapted from ref. [23]. **b** Western blot for one of the three biological replicates from independent clones is shown. Data are quantified in Supplementary Fig. 10. Upper panel: probed with anti-GFP to visualize the reporter. Lower panel: anti-Pgk1 loading control. Migration positions are indicated for the stalled peptide (stalled) and for this peptide modified by addition of a CAT-tail (CAT). Also indicated are the major product from read-through of the stalling sequence (read-through) and the full-length protein (FL). The region around the FL product is shown in higher contrast in Supplementary Fig. 10a and the region containing stalled and CAT products in Supplementary Fig. 10c. **c** Northern blot analysis of stalling and non-stalling reporter transcripts in *ski2Δ* background using probe Np_GFP and loading control using probe Np_scR1. Autoradiograph for one of the three biological replicates from independent clones is shown. Data are quantified in Supplementary Fig. 11. Migration positions are indicated for the full-length mRNA (FL), the expected 5' product of mRNA cleavage at the stall (5' fragment/5'-frag.) and for heterogeneous, 5' truncated RNAs (truncated)

**Hel2-binding sites across mRNAs**. To assess whether Hel2 binds to translationally engaged mRNAs, we compared the recovery of each mRNA species in Hel2 CRAC data sets with either total mRNA abundance (RNASeq)[42] or with ribosome association (RiboSeq)[43] (Fig. 8a, b). Hel2 crosslinking was better correlated with ribosome association than with mRNA abundance, consistent with binding to mRNAs during translation. Preferential binding of Hel2 to highly translated mRNAs was supported by sorting mRNAs by Hel2 binding and comparing this to the ratio of sequence reads in RiboSeq versus RNASeq (Fig. 8b).

To localize Hel2 binding within mRNAs, we generated metagenomic plots of total read density and micro-deletion density (proxy for crosslink position) over the most strongly and reproducibly bound protein-coding genes for Hel2 CRAC and RiboSeq footprint data. CRAC reads only were aligned to the start codon (Start) or stop codon (Stop) (Fig. 8c–e; Supplementary Figs. 12, 13 for additional analyses). These analyses revealed that Hel2 binds mRNA predominantly at sequences flanking the translation stop codon, with lower read density over the actual stop codon position. The width of the valley in the metagene plot (~30 nt) is consistent with typical ribosome footprints (Fig. 8d); however, individual sequences showed variations in size (Supplementary Fig. 12,j–l). In contrast, RiboSeq footprints that reflect protection by ribosome peak close to the stop codon position (Fig. 8g).

This indicates that Hel2 predominantly binds to mRNA at sites not protected within the terminating ribosome. Binding was strongest 3' to the stop codon, with deletions peaking 10 nt upstream of the pA site (Supplementary Figs. 12c and 13f). This would be consistent with ribosome-bound Hel2 preferentially contacting mRNAs close to the entry channel, in agreement with the 18S rRNA crosslinking data.

Hel2 CRAC read densities were strikingly low at the 5' termini of mRNAs (Fig. 8c), with increased binding commencing ~150 nt from the start codon (~270 nt from the transcription start;

Supplementary Fig. 12a). This distribution is in marked contrast to ribosome density, which is the highest close to the start codon (Fig. 8f). However, the 5'-terminal region with low Hel2 binding correlates well with the reported minimum separation between start codon and stalling sequences, required to trigger NGD[44].

The frequency of micro-deletions in the sequence data generally followed the binding profile. (Fig. 8d, e, Supplementary Fig. 13). Non-templated, oligo(A) tracts were identified at the majority (61%) of annotated polyadenylation sites recovered with Hel2, consistent with binding to mature mRNA (Supplementary Figs. 12e, f, 13g–i). Annotated sites lacking oligo(A) in the sequence data may result from alternative polyadenylation sites, which are common in yeast. In contrast, the level of oligo(A) tracts mapped within the transcript was very low and comparable to background levels (Supplementary Fig. 12d). Known targets of nonsense-mediated decay, including pre-mRNAs with retained introns, were not enriched and showed no specific Hel2-binding pattern.

Together, the data support a model in which Hel2 is associated with the 40S subunit via binding to 18S rRNA and can contact translating mRNAs at both the entry and exit channels, with greater interactions around the entry site.

**Loss of 18S interactions alters mRNA binding by Hel2**. To assess the links between 18S rRNA binding and mRNA interactions, we determined the mRNA read distribution of Hel2$_{1-492}$, Hel2$_{1-543}$ and Hel2-HTP, *asc1Δ* (Fig. 9, Supplementary Figs. 14, 15).

Both Hel2$_{1-492}$ and Hel2-HTP, *asc1Δ* showed decreased mRNA reads as a fraction of total reads, despite the reduced 18S rRNA interactions (Supplementary Fig. 14). Conversely, relative tRNA binding was increased in the truncation mutants, across all tRNA species (Supplementary Fig. 14). The sequence data are normalized to total reads, so the apparent increase in

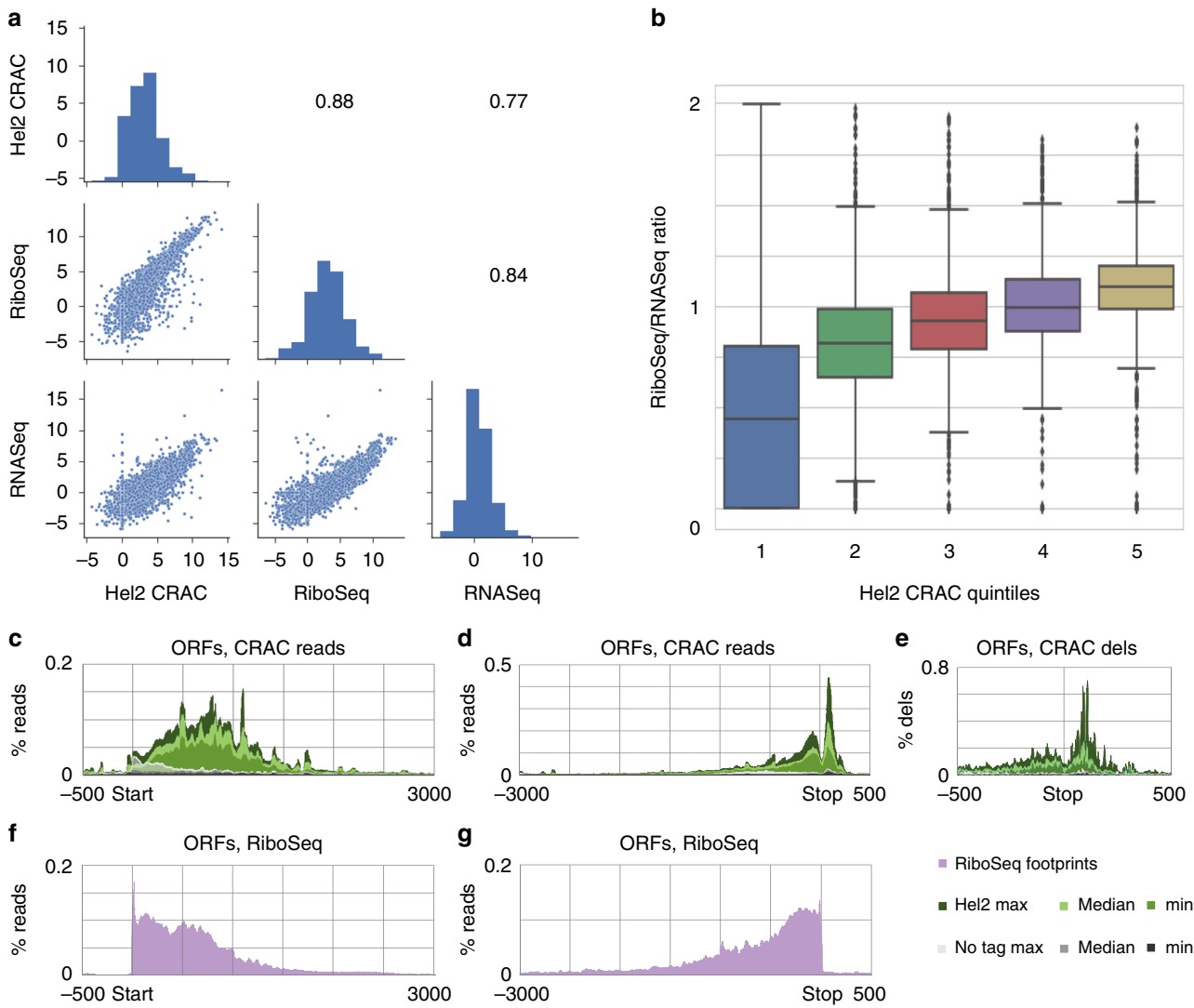

**Fig. 8** mRNA binding by Hel2. **a** Abundance of protein-coding genes in Hel2 crosslinking and analysis of cDNA (CRAC) data sets (median of 6 data sets for each gene) compared to RiboSeq (1 data set; from ref. [43]) and RNASeq (median of 3 data sets; from ref. [42]). The latter two were from isogenic wild-type strains. All axes are in log2. Spearman correlation between abundances of protein-coding genes in RNASeq, Hel2 CRAC and RiboSeq (medians) is shown in the right panels. **b** Genes were grouped into quintiles according to Hel2 CRAC read counts, in hits per million per kilobase. These are plotted against the ratio of RiboSeq reads and RNASeq reads for each gene in the quintiles. Centre lines of box plots show the medians; box limits indicate the 25th and 75th percentiles; whiskers extend 1.5 times the interquartile range from the 25th and 75th percentiles, data points are plotted for outliers. **c–e** Metagene analysis of collapsed data sets performed for the subset of top 275 reproducibly Hel2-bound protein-coding genes with mRNA length of ≥500 nt. **c, d** Total reads mapped across open reading frames (ORFs) aligned by the translation start or stop codons. **e** Micro-deletions mapped across stop codons (smoothed data shown here, unsmoothed data shown in Supplementary Fig. 13). **f, g** RiboSeq footprints mapped across ORFs aligned by the translation start or stop codons. The values for "% reads" or "% deletions" shown are normalized to the sum of total Hel2 and no tag CRAC signal (**c–e**) or to the sum of all RiboSeq footprints (**f, g**) over the top reproducibly bound genes

tRNA association may reflect reduced 18S and mRNA reads. However, RNA recovery in the Hel2$_{1-492}$ and Hel2-HTP, asc1Δ strains was clearly above the background in untagged strains. This indicates that additional domains contribute to mRNA and tRNA binding, in addition to the sites identified by iRAP.

In mRNA metagene analyses, the notable increase in signal seen after +200 nt in the wild-type was almost absent for Hel2$_{1-492}$, Hel2$_{1-543}$ and Hel2-HTP, asc1Δ (Fig. 9, Supplementary Fig. 15). The strong peaks flanking the stop codon were also greatly reduced. In Hel2$_{1-543}$, the reduction in the upstream peak

was less marked, whereas the downstream peak was most reduced. This was, however, variable between experiments.

These observations are consistent with the model that the distinctive distribution of Hel2 across mRNAs reflects their interactions with ribosome-bound Hel2. In the truncation and asc1Δ mutant strains, the residual association of Hel2 with mRNAs may reflect a different recruitment pathway that results in a much lower, but more even, distribution across mRNAs.

A small number of mRNAs showed increased crosslinking to Hel2 in the mutant strains, including asc1Δ (Supplementary

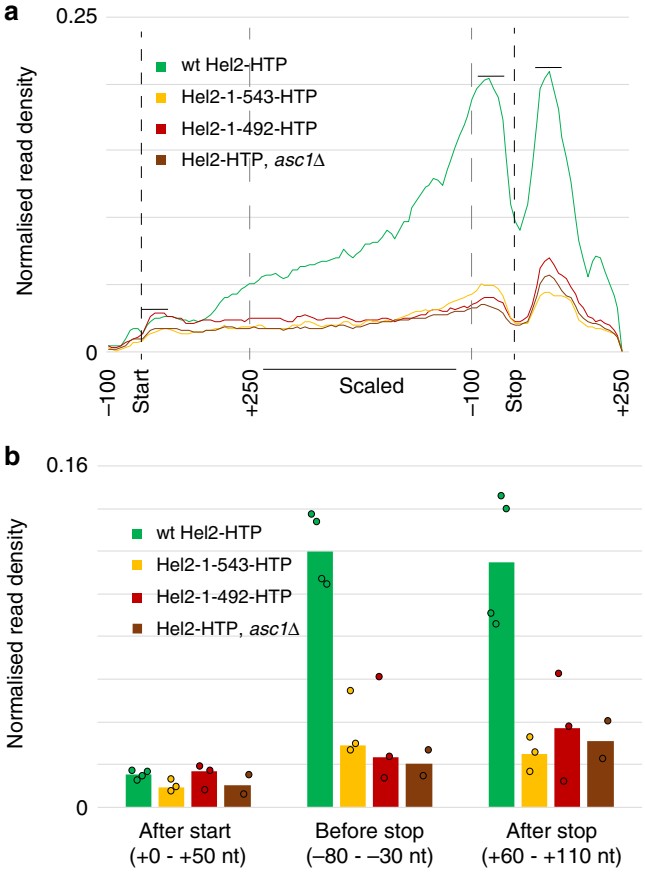

**Fig. 9** Effect of Hel2 mutations and *asc1Δ* on Hel2 mRNA crosslinking. Effects determined for top 252 reproducibly Hel2-bound mRNAs of transcript length >500 nt and open reading frame length >350 nt.
**a** Metagene representation of read density over mRNAs from Start −100 nt to Stop +250 nt with nucleotides between start codon +250 nt and stop codon −100 nt (inner dashed lines) scaled. Data were separated into bins of 10 nt for non-scaled data and an additional 50 bins for scaled regions. For each data set, and for each gene, data were normalized to total read density over the gene. After summing up normalized data for each data set, read counts were then scaled according to the ratio of total read density of the respective data set and corresponding wild-type data set. The sum under the curve was set to 1 for wild-type and the other curves were scaled accordingly. Plots represent medians of normalized, scaled data from all available data sets. mRNAs were scaled between start codon +250 nt and stop codon −100 nt. Horizontal lines indicate where data were summed to give values for quantitation shown in **b**. **b** Sums of read densities after start codon (60 nt/6bins after Start codon), before stop codon (−90 to −30 nt/ 6 bins) and after stop codon (+50 to +110 nt/6 bins). Bars represent medians; circles represent data points from separate biological replicates

Fig. 16a), which was apparently independent of changes in mRNA abundance or translation efficiency (Supplementary Fig. 16b, c). These included TY retrotransposon transcripts, mRNAs encoding Hac1, Ire1, Cho2 and 5' exonuclease Rat1. The distribution of Hel2 was analysed across *HAC1* (Supplementary Fig. 16a). On this mRNA, the distribution of wild-type Hel2, Hel2$_{1-492}$ and Hel2 in the *asc1Δ* background was strikingly different from the metagene analysis. Binding of Hel2 was enriched at the 5' end, in contrast to the depletion seen on most mRNAs, and this was increased by mutations, Hel2$_{1-492}$ and *asc1Δ* that impair 18S rRNA association. We speculate that these

RNAs are preferentially bound by Hel2 in the absence of its 18S association, using the alternative recruitment pathway.

## Discussion

Our data are complementary to previous studies reporting Hel2/ hZNF598 ubiquitination activity and substrate specificity[10–12,22]. Ribosomal association of Hel2/hZNF598 had previously been observed[22,24]. We were now able to pinpoint the molecular interactions between Hel2 and its ribosomal RNA-binding partners, at single amino acid and nucleotide precision.

UV crosslinking identified two adjacent amino acids in the CTD as crosslinked to RNA (Fig. 1)[31]. Complementary mapping data for protein-binding sites on RNA confirmed this RNA-binding activity (Fig. 2), revealing that Hel2 binds both the small subunit of translating ribosomes (Fig. 3) and associated mRNAs (Fig. 8).

The major and one minor 18S rRNA crosslinking sites (C1490-A1492; U1494) are located close to the mRNA entrance channel, while a second minor crosslink (U1362) lies close to the mRNA exit channel (Fig. 3). Consistent with this, Rps3 and Rps20, which are ubiquitinated by Hel2, are located at the entrance channel. Indeed, Rps3 directly contacts a nucleotide crosslinked to Hel2 in the major rRNA interaction site. hZNF598 also ubiquitinates hRPS3 and hRPS20 but has most activity against hRPS10, similarly located close to the entry channel[10–12]. Rps10 from *S. cerevisiae* and related yeasts lack the C-terminal region including the ubiquitinated lysine residues (Supplementary Fig. 5). This correlates with changes in the pattern of Hel2 conservation, suggesting coevolution of Hel2 and its ubiquitination targets. Binding of Hel2 to 18S rRNA was largely lost in the absence of the ribosome-associated protein Asc1 (homologous to hRACK1) (Fig. 5, Supplementary Fig. 7). The major Hel2 crosslinks in 18S rRNA are positioned ~62 Å from Asc1 (Fig. 3), consistent with physical interactions, and the lysine residue ubiquitinated by Hel2 in Rps3 (K212) is also close to Asc1 (~15 Å). In collided di-ribosomes, the interface between both 40S subunits contains hRACK1 of the first (stalled) ribosome and hRPS10, hRPS20 and hRPS3 of the following (collided) ribosome[45] and hRPS10 ubiquitination by hZNF598 happens in the context of the collided ribosomes[45]. The major Hel2 crosslink site (C1490-A1492 in yeast; C1554-A1556 in human) and the minor crosslink site of the 3' domain (U1362 in yeast; U1426 in human) both lie close to the 40S–40S interface, indicating that Hel2 and Asc1 may be even closer than judged from the monosome structure. However, the second minor crosslink site (U494 in yeast; U542 in human) would be buried between the collided ribosomes. Differential accessibility may help Hel2/hZNF598 distinguish between single stalled ribosomes and di-ribosomes.

A C-terminal truncation in Hel2$_{1-492}$, which removed the crosslinked amino acids (L$_{501}$ and K$_{502}$), almost completely abolished 18S rRNA crosslinking at the major site (Fig. 4), and this was associated with the loss of Hel2 recovery in polysomes (Fig. 6). In a reporter assay, losses of RQC and NGD in Hel2$_{1-492}$ strains were similar to losses in *hel2Δ* (Fig. 7, Supplementary Figs. 10, 11). This was also reflected by comparable chemical sensitivity (Supplementary Fig. 8). A shorter C-terminal truncation, Hel2$_{1-543}$, that retained the crosslinked amino acids showed intermediate phenotypes in all these respects. These results demonstrated a direct requirement for Hel2 in NGD. They further suggest that association of Hel2 with the 40S subunit within polysomes, modulated through 18S rRNA binding, is required for both RQC and NGD. In contrast, Hel2$_{LKAA}$-HTP and Hel2$_{LKDD}$-HTP crosslinked to rRNA with similar efficiency to Hel2-HTP

(Fig. 4) demonstrating that mutations of $L_{501}$ and $K_{502}$ are not sufficient to abrogate the Hel2–RNA interactions. Consistent with this conclusion, Hel2 from *C. glabrata* contains a glycine residue at the position corresponding to $K_{502}$ (Fig. 1c). However, the chemical sensitivity screens (Supplementary Fig. 8) show that the point mutations do modulate Hel2 activity.

While the functional importance of Hel2 mRNA binding is unclear, our mRNA crosslinking data sheds light on the distribution of Hel2-associated ribosomes between mRNA species and along mRNAs. The level of Hel2 crosslinking to mRNA species was strikingly well correlated with RiboSeq data, which reflects ribosome occupancy (Fig. 8a), but less correlated with RNASeq, reflecting mRNA abundance. We had anticipated that Hel2 would be preferentially recruited to mRNAs associated with aberrantly stalled ribosomes and correlated with features like codon bias. However, these data indicate that Hel2 binds all translationally engaged mRNAs, with little variation.

The distribution of Hel2-binding sites across mRNAs was consistent between most mRNA species (Fig. 8). It was, however, strikingly uneven along mRNAs, with strong peaks flanking the stop codon (Fig. 8). These peaks were greatly reduced by the Hel2$_{1-492}$ truncation and in *asc1Δ* (Fig. 9, Supplementary Fig. 15) indicating that they involve 18S-associated Hel2. We postulate that surveillance activity predominately involves 40S-associated Hel2 scanning all translation termination complexes for aberrant features. This model would be in agreement with findings of increased ribosome densities in 3' untranslated regions (UTRs) in the absence of splitting factors Dom34/Hbs1[46] and Rli1[47], which function downstream of Hel2 in RQC.

In contrast, Hel2 binding was very low over the initial ~150 nt of most ORFs (Fig. 8c). Notably, ribosome occupancy is high over the first ~200 codons[48,49] with a high prevalence of stacked ribosomes, especially over the first ~25 codons[50]. However, NGD fails to operate in the start codon-proximal region[44], which may be important to avoid premature degradation of stacked ribosome–mRNA complexes. We speculate that translation initiation factors (TIFs) occlude the major 18S rRNA-binding site for Hel2 and that this contributes to protection during early elongation or that factors interacting with nascent peptides, which protrude from the ribosome only after ~30-40 codons[51], are needed to recruit Hel2. Immunoprecipitation of Hel2[23] showed enrichment for nascent chain-interacting factors Nat1 and Zuo1 and for TIFs, which potentially occlude or promote Hel2 recruitment. These included components of eIF3, which has binding sites adjacent to hRACK1 in humans[52] and Asc1 in yeast, and subunits of eIF3 also interact physically with Hel2[53] and Asc1[54].

Notably, crosslinking of Hel2 to the 5' region of mRNAs was not clearly affected by truncation of the RNA-binding region in Hel2$_{1-492}$ or loss of Asc1. In these strains, recovery of mRNAs with Hel2-HTP also remained greater than in the non-tagged control (Supplementary Fig. 14, Supplementary Tables 6–7) supporting the existence of an alternative pathway for Hel2 recruitment to mRNA. A small number of mRNAs appear to be preferentially targeted (Supplementary Fig. 16), since they showed increased relative binding in the Hel2$_{1-492}$ and *asc1Δ* strains. Interestingly, these included HAC1, whose expression is controlled by differential translation of mRNA isoforms generated by non-canonical splicing, and IRE1, encoding the regulator of HAC1 splicing. Consistent with an alternative recruitment pathway, the distribution of Hel2 across HAC1 was strikingly different from most other mRNAs, with high 5' binding and no obvious peaks flanking the stop codon (Supplementary Fig. 16a).

Recent PAR-CLIP crosslinking data[12] also place hZNF598 on translating ribosomes and indicate roles for the CTD in RNA binding and RQC. Differences in the details of the interactions recovered (preference of tRNA over mRNA and rRNA; different, less specific ribosomal peaks; flat distribution over mRNAs and preferential enrichment of RQC-triggering tRNAs) may be technical (CRAC in yeast vs. PAR-CLIP in human cells) or may reflect the coevolution of the CTDs of Hel2 and hZNF598 and their ubiquitin ligase targets. Overall, the good agreement of our data with published data on Hel2/hZNF598 targeting[10–12,22] and NGD/hZNF598 target recognition[44,45] give us confidence that our data are relevant for both yeast and mammalian surveillance.

In conclusion, the data presented support the model that Hel2 associates with 40S ribosomal subunits and scans ribosome–mRNA complexes, particularly during translation termination. The 5' regions of most mRNAs escape this mechanism, possibly to avoid inappropriate targeting of slow or stacked ribosomes for RQC in early translation. A small number of mRNAs apparently recruit Hel2 via a different pathway, which may be linked to translation regulation.

## Methods

**Strains and oligonucleotides.** Strains used in this work are listed in Supplementary Table 8. Oligonucleotides used are listed in Supplementary Table 9.

**Construction of strains.** Hel2-HTP, Hel2$_{1-492}$-HTP and Hel2$_{1-543}$-HTP were generated by PCR-amplifying the HTP-tagging cassette from pBS1539-HTP[32] with respective primers, transforming BY4741 with the PCR product and selection on –URA media.

Hel2$_{1-492}$ and Hel2$_{1-543}$ were generated by transforming Hel2–HTP with bridging dsDNA created by primer extension, with homology to regions upstream and downstream of the region to be deleted, and selection on 5-FOA media. Hel2$_{LKAA}$/$_{LKDD}$ and the respective HTP-tagged versions were generated by amplifying the region downstream of the mutation with the respective mutation primers and downstream primers from either BY4741 or Hel2-HTP, extending the mutated dsDNA using extension primer and downstream primer, transforming the PCR product into Hel2$_{1-492}$-HTP for non-tagged or Hel2$_{1-492}$ for tagged versions and selecting on 5-FOA media for untagged or on –URA media for tagged versions. The *hel2Δ* (delitto perfetto) strain was generated by a modified version of the 50:50 method[55]. To create a pop-out construct, the HTP-tagging cassette was amplified from pBS1539-HTP with respective primers, transformed into BY4741, selected first on –URA media for integration of the pop-out-HTP-tagging cassette construct, cultured in YPEG media for 2 days at 30 °C and then selected on 5-FOA media to identify pop-out. *hel2Δ::KanMX*; *asc1Δ* and Hel2-HTP, *asc1Δ*; as well as all *ltn1Δ* and *ski2Δ* strains, were generated by PCR-amplifying the KanMX cassette from pFA6a-kanMX6 with respective primers, transforming respective strains with the respective PCR product and selection on G418-containing media. All strains were verified by PCR amplification with flanking primers and/or primers annealing to the tag/cassette. Point mutants were additionally verified by Sanger sequencing.

Strains carrying reporter plasmids were transformed with pGTRR[23] (containing a 12 arginine stalling sequence) or pGTSTR[23] (containing a non-stalling sequence) and were selected on –URA media.

**Crosslinking and analysis of cDNA.** For CRAC[56], yeast strains were grown in -TRP SD media containing 2% glucose to OD$_{600}$ 0.5, crosslinked for 100 s and harvested by centrifugation. Cells were lysed in lysis buffer (50 mM Tris-HCl, pH 7.8, 150 mM NaCl, 0.1% Nonidet-P-40 substituent (Roche), 5 mM 2-mercaptoethanol) containing complete protease inhibitor, using zirconia beads, and cleared lysates were incubated with IgG sepharose. After vigorous washing in high-salt buffer (like lysis buffer but 1 M NaCl) and lysis buffer, bound proteins were cleaved from the solid support using TEV protease, RNA was subjected to mild digestion (RNaceIT, 1:100 diluted, 5 min) and digest was stopped by addition of a high concentration of guanidinium hydrochloride (final concentration 6 M). The resulting mixture was incubated with Ni-NTA sepharose beads overnight, followed by extensive washes with wash buffer I (50 mM Tris-HCl, pH 7.8, 300 mM NaCl, 0.1% Nonidet-P-40 substituent, 10 mM imidazole, 5 mM 2-mercaptoethanol, 6 M guanidinium hydrochloride). Dephosphorylation, 3'-linker ligation (0.5 U/μl T4 RNA ligase 1, 5 U/μl T4 RNA ligase 2, tr, KQ), re-phosphorylation with γ-$^{32}$P-ATP and non-labelled ATP and 5'-linker ligation were then performed on the Ni-NTA, always in the presence of 1 U/μl RNaseIN RNase inhibitor. Following washes with wash buffer II (50 mM Tris-HCl, pH 7.8, 50 mM NaCl, 0.1% Nonidet-P-40 substituent, 10 mM imidazole, 5 mM 2-mercaptoethanol) protein–RNA complexes were then eluted in elution buffer (wash buffer II + 250 mM imidazole), precipitated with TCA or acetone, size-separated on NuPAGE 4-12% Bis–Tris gels, transferred to nitrocellulose membranes, visualized by autoradiography and appropriately sized complexes were excised: size of Hel2 + crosslinked RNA was excised in the first set of experiments, while a region of the size of Hel2-1-492 + crosslinked RNA to the size of wild-type Hel2 + crosslinked RNA was excised in

the second set of experiments where mutants were analysed. Membrane slices of equal size were pooled within each experiment and further processed as combined samples. Complexes were released from the membrane and proteins were degraded by digestion with Proteinase K; the resulting RNA was purified by phenol–chloroform–isoamyl alcohol extraction and ethanol precipitation. RT was performed with indexed primers (Supplementary Table 9); primers were digested with Exonuclease I (Thermo Scientific, 2 × 30 min with addition of 40 U ExoI before each incubation, followed by heat denaturation at 80 °C for 10 min). Twenty-one cycles of PCR were performed and libraries purified (QIAQuick PCR purification kit, Qiagen), size selected on 2.5% Metaphore agarose (Lonza) and gel-extracted using the MinElute Gel Extraction Kit (Qiagen). After quantification (Qubit, Thermo Scientific), libraries in the first set of experiments (wild-type Hel2-HTP) were pooled and sequenced in a single run on HiSeq2500 (Edinburgh Genomics), in fast mode, on two lanes, with 100 bp read length, using custom primers to read out the indices. Libraries of the second set of experiments were sequenced in three runs (exp.1, exp. 2, exp. 3+4) on a MiniSeq system, with 75 bp read length.

**Western blot analysis of CRAC purification**. To quantify the amounts of different versions of Hel2-HTP over the CRAC purification procedure, samples were electrophoresed on NuPAGE 4–12% Bis–Tris gels and transferred onto iBlot2 nitrocellulose membranes using the iBlot2 system. Membranes were blocked with 5% milk in phosphate-buffered saline (PBS)-Tween for 1 h, incubated with rabbit anti-TAP antibody (Thermo Scientific CAB1001, 1:5000) overnight at 4 °C, shaking, washed 3 × 20 min with PBS-T at room temperature, shaking, then incubated for 1 h with ECL Anti-Rabbit IgG (GE Healthcare NA934V, 1:5000) in 5% milk in PBS-T, shaking at room temperature. After washing, membranes were incubated for ~ 3 min with ECL and then imaged in a Chemidoc Touch system (BIORAD). Input blots were then stripped, and membranes were blocked again and incubated with mouse anti-Pgk1 antibody (Thermo Fisher PA528612, 1:2000), as primary antibody and anti-mouse IgG (GE Healthcare NXA931, 1:5000) as secondary antibody and imaged as described above. ImageJ[57] was used for quantification of signals. Rectangles of equal area were used to quantify bands and background for background subtraction.

**Sequence data analysis**. i7-indexed sequences were demultiplexed using the Illumina software. For samples sequenced on two separate lanes, fastq files were concatenated prior to further processing. Treating each index separately, index-separated files were then barcode-separated by the in-line barcodes used in library preparation, using pyBarcodeFilter.py from pyCRAC[58]. In cases where the in-line barcodes ended with a randomized nucleotide and in cases where RTs were carried out with >1 separate indexed primer from the same sample, this resulted in separate fastq files, treated separately in the following steps. Using flexbar[59] (-f (format) i1.8, -n (threads) 3, -ao (adapter minimal overlap) 4, -q (pre-trim-phred) 30, -u (max. uncalled) 3, -g (removal-tags)), sequences were trimmed and quality filtered. Using pyFastqDuplicateRemover.py, fastq files were then collapsed (separately), taking into account the three random nucleotides contained in the in-line barcodes. After collapsing, resulting fasta files from the same sample were concatenated (cat, for analysis of "collapsed" sequences), as were flexbar output fastq files (for analysis of "raw" sequences). Sequences were then aligned to a S. cerevisiae genomic reference file based on version EF4 (Ensembl release 74)[60], using novoalign (-c (threads) 55, -s (shorten by this number of bases if not aligned) 1, -r (method) Random, -l (minimum length) 17). Novoalign output files were used for several further analyses. First, hittables were produced from both "raw" and "collapsed" data, using pyReadCounters.py and a gene feature file that contained a range of coding and non-coding features. Hittables were employed to compare the abundance of specific RNAs as well as RNA classes between data sets. The output of pyReadCounters.py was also used to generate bedgraph genome browser tracks with pyGTF2bedGraph.py. Third, pyPileup.py was used to calculate numbers of reads, substitutions and deletions over the RDN37-1 gene from "raw" and over the TDH3, TPI1 and CCW12 genes from "collapsed" novoalign output files. To compare different samples, reads, substitutions and deletions were normalized (sum of all values = 1) and maximum, mean, and minimum were plotted over the gene. For CRAC involving mutants, a number of samples had been processed as two separate size fractions. For those cases, numbers from both fractions were summed up prior to normalization.

To generate metagene plots, a custom script, 2Dplotter, was used, using reference files containing the ~90% of the top 300 reproducible Hel2-bound transcripts with lengths of at least 500 nt (275 genes), using either start and stop codon or transcription start site and polyadenylation site (pA) for alignment. Summing and normalization were performed as described above.

For metagene analysis of deletions, bedgraphs were generated using the --count option of pyGTF2bedGraph and -t deletions to only count deletions. Bedgraph files of plus and minus strand were converted to bigwig format using bedGraphToBigWig and a single, non-strand-specific wig file was generated using the sum option of wiggletools[61]. Wig files were converted back to bigwig format using wigToBigWig and used as input files for the deepTools[62] module computeMatrix in reference-point mode and a bin-size of 1 nt (-bs 1). Individual gene profiles were then calculated with deepTools module plotProfile for the same

subset of 275 reproducibly bound transcripts of at least 500 nt length. Data were plotted either as raw data or after applying 9 nt rolling average smoothing.

For metagene plots and two-dimensional (2D) plots that were scaled over part of the gene body, data were processed similarly but without the –t option in pyGTF2bedGraph. To generate metagene plots, all sub-data sets of each sample were combined using sum and converted back to bigwig format. computeMatrix was used in scale-region mode, with the following setting: -b 100 -a 250 --unscaled5prime 250 --unscaled3prime 100 --regionBodyLength 500, with the default bin-size of 10 nt and data were then further processed to calculate total number of reads over all 275 genes (of which 252 had coding sequences long enough to be analysed.) To generate metagene plots, sums over each gene were calculated and each gene's reads were normalized to that number. Normalized reads were summed up for each position and normalized for each sample to the ratio of total reads in the respective sample and total reads in the corresponding wild-type Hel2-HTP sample. Before plotting, the area under the curve was then set to 1 for wild-type and accordingly for the other samples. Alternatively, data were not normalized to the ratio of total reads in sample to wild-type Hel2-HTP but the area under the curve was set to 1 for all samples.

For 2D plots, summed data sets for each sample were normalized to the number of total reads (from metagene plot calculations) over all 275 genes in the corresponding wild-type Hel2-HTP data set. Following this normalization, all data sets corresponding to the same strain background were summed and divided by the number of data sets. Resulting files were converted back to bigwig format and processed as described for metagene plots. plotHeatmap was then used to generate the 2D plots.

Poly(A) analysis[63] was performed on reads containing the 3'-adapter using a custom script. From reads identified to contain a non-encoded oligo(A) tail, the tail, as well as any trailing As were removed. Resulting adapter-containing and A-tail removed fasta files were subjected to alignment and metagene analyses for reads and deletions and plotted as described above. To determine the percentages of oligo-adenylated reads at the pA site, the numbers of reads from the metagene total at the pA site in oligo-adenylated reads were divided by the numbers of reads in adapter-containing reads.

To compare Hel2 binding to RNA-seq[42] and ribosome profiling (RiboSeq)[43], data sequencing reads were mapped to a database of mature transcripts. The database was created by addition of the 5' and 3' UTRs[64], excising introns and arbitrary addition of 10 adenine nucleotides at the 3' end to mimic the poly(A) tail of mature mRNAs. Coverage of reads mapped to each mature mRNA was normalized to 1 kb per million and median value for each type of experiment (RNASeq, RiboSeq or Hel2 CRAC) was used for comparison. Median value was log2 transformed to reduce the effect of outliers. Spearman correlation coefficients were calculated for median values. Finally, Hel2 CRAC data were divided into five quintiles and RiboSeq-to-RNASeq ratio was calculated for genes included in each quintile. For other analyses, alignment to the genome was preferred, as this accounts for use of alternative transcription start or polyadenylation sites, as well as for included introns (e.g. for HAC1).

**Ribosome and polysome association of Hel2**. Analyses were performed similar to a recent publication[22]. Overnight cultures of strains containing wild-type or mutant Hel2-HTP in wild-type or asc1Δ background were grown in YPDA. Cultures (100 ml) were diluted to OD600 of 0.1 and grown to an OD600 of 0.8 at 30 °C. Cycloheximide was then added to 0.1 mg/ml final concentration, cultures were shaken at 200 rpm, at 30 °C for 5 min, then cells were pelleted by centrifugation at 4000 rpm, 2 × 1 min at 4 °C, washed once with 20 ml cold H2O, before being snap-frozen. For lysis, cell pellets were dissolved in 200 μl lysis buffer (20 mM HEPES-KOH, pH 7.4, 100 mM potassium acetate, 2 mM magnesium acetate)[22], transferred to screw cap tubes containing 200 μl zirconia beads. Cells were then lysed in two consecutive runs on a Fast Prep-24 (MP Biomedicals) using the Cool Prep adapter, pre-cooled before being used and cooled during each run with 8 g dry ice at the following settings: 40 s, 6 m/s; or by vortexing for 10 min at 4 °C. After mixing with 200 μl lysis buffer, lysates were cleared twice by centrifugation at 13,000 rpm in a cooled microcentrifuge at 4 °C and OD260 was measured on a NanoDrop. An equivalent of 20 OD260 was loaded onto 10–45% sucrose gradients in 1× gradient buffer (10 mM Tris-acetate, pH 7.4, 70 mM ammonium acetate, 4 mM magnesium acetate)[22] prepared using a Gradient Master (BioComp). Gradients were centrifuged in an SW40-Ti rotor in an Optima XPN-100 Ultracentrifuge (Beckman Coulter) at 38,000 rpm, 4 °C for 2.5 h, (acceleration 1, deceleration 9). Gradients were then fractionated into 60 fractions of 200 μl using a Piston Gradient Fractionator (BioComp), equipped with a TRIAX flow cell (BioComp) for UV profiling and an FC203B fraction collector (Gilson). Fractions belonging to the same peak were pooled and TCA precipitated by addition of 20% (final) TCA and 20 min centrifugation at 13,000 rpm, 4 °C. Pellets were washed once with 20% TCA and once with 100% acetone, dried briefly and re-dissolved in 60 μl (first peak only) or 30 μl 1× NuPAGE LDS loading buffer, heated to 100 °C for 5 min and frozen. Prior to loading equivalent amounts of each peak's solution on NuPAGE 4–12% Bis–Tris protein gels (1 mm, 10 wells), samples were again heated to 100 °C for 5 min and centrifuged briefly. Gels were run 15 min at 50 V, then at 120 V and transferred using the iBlot2 system. Membranes were blocked with 5% milk/PBS-Tween for at least 1 h and probed overnight with peroxidase–anti-peroxidase antibody (Sigma Aldrich P1291, 1:2000) at 4 °C, shaking, washed and visualized as described above. Blots were then stripped, membranes were blocked again and blotted with mouse

anti-Pgk1 antibody (Thermo Fisher PA528612, 1:5000), as described above. ImageJ[57] was used for quantification of signals. Whenever possible, rectangles or other quadrangles of equal area were used to quantify bands and background for background subtraction and mean intensity values were used for relative quantification. If not possible, rectangles or quadrangles of different sizes were used, and the product of volume and mean intensity was used for relative quantification after background subtraction.

For quantification of $A_{260}$ profiles, profiles were aligned to start positions (to cancel dead volume), then normalized to 1 (area under the curve) and smoothed by 5-point sliding window median filtering. Smoothed data were plotted and used for quantification of peaks (sum of all values belonging to each peak).

**Quantitative PCR (qPCR).** For RT-qPCR experiments with total mRNA, overnight cultures from single colonies were diluted to $OD_{600}$ 0.1 and cultured to $OD_{600}$ 0.5. Cultures were harvested in 50 ml Falcon tubes upon addition of ~50% wet ice, by centrifugation, resuspended in ice-cold PBS and transferred to Eppendorf tubes and pulsed up to maximum speed for 1 s, then stored at −80 °C. RNA from different cultures was purified by a small-scale version of a previously published protocol[65] and ethanol precipitated, re-dissolved in 40–50 μl $H_2O$ and quantified (Nanodrop). For RT-qPCR experiments with sucrose-fractionated RNA, fractions from sucrose gradients (see above) for wild-type Hel2-HTP or $Hel2_{1-492}$-HTP were pooled into three groups: polysomal, monosomal, and earlier (free). In all, 5% of total pooled fractions were brought to 250 μl by addition of $H_2O$ and were then added to 250 μg of guanidinium thiocyanate, 10.6 μl of 0.5 M EDTA (pH 8), 5.25 μl 2-mercaptoethanol, 52.75 μl 20% sarkosyl and 500 μl phenol, vortexed and incubated at 65 °C for 10 min, then on ice for 10 min. Purification was then continued as above, and RNA pellets were re-dissolved in 20 μl $H_2O$ and quantified by Nanodrop.

Ten μg of total RNA or the complete sucrose fraction (all <10 μg) were DNase treated with Turbo DNase (Thermo) according to the manufacturer's protocol, including inhibition using the provided inhibitor beads. RNA was then ethanol precipitated, re-dissolved in $H_2O$ and quantified and 2 μg were subjected to RT using either the RetroScript Kit (Thermo) according to the manufacturer's protocol (total mRNA) or SuperscriptIII (sucrose fractions), using random decamers in both cases. cDNA was diluted 10-fold and 2 μl of this dilution were used in 5 μl scale qPCR in a LightCycler480 system (Roche), using the SYBR Premix Ex Taq II (Tli RNase H Plus; Takara) and the following protocol: initial denaturation: 95 °C, 30 s, 20 °C/s; 40 cycles of: denaturation: 95 °C, 5 s, 20 °C/s, annealing and extension: 60 °C, 20 s, 20 °C/s; melting curve analysis: 95 °C, 0 s hold, 20 °C/s, 65 °C, 15 s hold, 20 °C/s, 95 °C 0 s hold, 0.1 °C/s. For each sample, a no-RT controls were performed to ensure that levels of genomic DNA contamination were negligible, and for each primer combination in each run of qPCR, a standard was performed with a dilution series of genomic DNA and amplification factors determined therein were used for relative quantification. Relative changes in total mRNA levels of target genes were then normalized to average relative changes of the two control transcripts RPL19B and scR1. For each strain (Hel2-HTP or $Hel2_{1-492}$-HTP), relative levels of mRNAs in the free, monosomal and polysomal fraction determined by qPCR were multiplied by relative levels of RNA recovery in each fraction to determine the actual percentage of each mRNA in each fraction.

**Growth tests.** For growth tests, overnight cultures from single colonies were diluted to 0.1 $OD_{600}$, cultured into exponential phase (roughly ~0.2–0.8 $OD_{600}$) and diluted to $OD_{600}$ 0.1. Fifty μl of diluted culture were then combined with 50 μl of complete S.D. media with or without 100 mM HU, 100 μg/ml AZC, 4 μg/ml anisomycin or 20 μg/ml anisomycin in 96-well plates. Plates were then sealed and placed into a Sunrise plate reader (Tecan), with continuous shaking, taking measurements every 15 min. For each sample, a technical duplicate was carried out. Numbers of biological replicates ( = number of data points) and number of clones used to generate those data points are given in the respective figures.

**Reporter assays.** To study the effect of Hel2 mutations on functionality of NGD and RQC, we generated strains that carry reporter plasmids encoding reporter constructs consisting of GFP and RFP on a single transcript, separated by a TEV protease site and a stalling sequence (12 arginine codons; pGTRR[23]) or a non-stalling sequence (6 repeats of serine-threonine; pGTSTR[23]). To test NGD, strains were generated in a *SKI2*-deleted background. –URA minimal media (CSM) was inoculated with the respective strain and grown at 30 °C, 200 rpm overnight. Cultures (20 ml) were diluted to $OD_{600}$ of 0.1 and grown to an $OD_{600}$ of 0.5. Fifteen ml were harvested, washed once with $H_2O$ and frozen at −20 °C. RNA was extracted and precipitated as described above for qPCR, diluted in 50 μl $H_2O$ and quantified by Nanodrop. Ten μg of RNA were mixed 1:1 with NorthernMax-Gly loading dye (Thermo Scientific), incubated for 30 min at 50 °C, loaded onto a 1.5% BPTE agarose gel and run at 40 V for ~16 h. RNA was partially digested in-gel with 75 mM NaOH for 20 min, then the gel was soaked in neutralization buffer (500 mM Tris-HCl, 150 mM NaCl, pH 7.5) for 20 min and transferred onto BrightStar-Plus (Ambion) membrane overnight. Blots were probed using ULTRAhyb-Oligo hybridization buffer (Ambion) and 5'-$^{32}$P-phosphorylated oligonucleotides probes against the GFP sequence to determine whether NGD has been initiated, were then

stripped and re-probed against scR1 mRNA as loading control. Quantification of FL band and scR1 control band were performed as described above for western blot analysis. Average and standard deviation were calculated from the three replicates, except for non-stalling $Hel2_{LKAA}$, where the third replicate was not taken into account due to low intensity. For analysis of signal distribution over the lane, a rectangle was placed over each lane in ImageJ and the plot profile function was used to determine signal distribution. Background was calculated as the average value from a lane not containing any bands and was subtracted from each value in the plot profile. Plot profiles were then aligned by the maximum value and normalized to the sum of all values. Profiles were then plotted as averages over all three samples.

To assess RQC, both $ltn1+$ and $ltn1\Delta$ strains were used. –URA minimal media (CSM) was inoculated with the respective strain and grown at 30 °C, 200 rpm overnight. Cultures (20 ml) were diluted to $OD_{600}$ of 0.1 and grown to exponential phase ($OD_{600}$ of 0.4–0.5) and harvested, washed once with $H_2O$ and snap-frozen prior to being stored at −20 °C. Pellets were lysed in a mixture of 100 μl of a 1 in 6 dilution of TCA, with 100 μl zirconia beads for 5 min at 4 °C. After addition of 900 μl of 5% TCA, samples were vortexed, and the supernatant centrifuged for 15 min, maximum speed, at 4 °C. Pellets were re-dissolved in 20 mM Tris pH8, and the protein concentrations were measured using Bio-Rad protein assay and a spectrophotometer. Equivalent amounts of sample, mixed 1 in 4 with NuPAGE 4× loading dye and heated to 100 °C for 5 min were loaded onto NuPAGE 4–12% Bis–Tris protein gels (1 mm, 12 wells). Gels were run and transferred, probed and imaged as described above for sucrose gradients, with the following differences: Membranes were probed overnight with mouse anti-GFP antibody (Roche, 11814460001, 400 μg/ml, 1:1000) as primary and with anti-mouse IgG (GE Healthcare NXA931, 1:5000) as secondary antibody. Blots were stripped and probed for Pgk1 as loading control, as described for sucrose gradients. Image quantification was performed as described for sucrose gradients, but all values measured for GFP signal were normalized to the Pgk1 signal in order to account for loading differences.

**Code availability.** Most of the code used in this work is available online. Custom code can be made available upon request through the corresponding author.

**Reporting summary.** Further information on experimental design is available in the Nature Research Reporting Summary linked to this article.

## Data availability

The data that support this work are available from the corresponding author upon reasonable request. The sequencing data have been deposited in NCBI's Gene Expression Omnibus and are accessible through GEO series accession number GSE114429. The source data underlying Figs. 2a, 3a, 4a, b, 5, 6a–c, 7b, c, 8a–g, 9a, b and Supplementary Figs. 4, 6b, 7, 8, 9, 10a–c, 11a–c, 12a–l, 13a–i, 14, 15a–c, 16a–c are provided as a Source Data file. A reporting summary for this Article is available as a Supplementary Information file.

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

## Acknowledgements

M.-L.W. was supported by an EMBO Long-Term Fellowship (ALTF 348-2015). The Wellcome Trust supported this work through a Senior Research Fellowship to J.R. (084229), a Principle Research Fellowship to D.T. (077248), a Centre core grant (092076)

and an instrument grant (091020), with additional support from the Wellcome Trust/ Edinburgh University via the Institutional Strategic Support Fund. L.P. was supported by a Marie Curie Intra European Fellowship within the 7th European Community Framework Programme and by an Estonian Research Council grant (PUT626). T.W.T. was supported by Polish Ministry of Science and Higher Education, Mobility Plus program (1069/MOB/2013/0). Reporter plasmids pGTRR and pGTSTR were a kind gift from Dr. Onn Brandman (Stanford University). The authors thank Dr. H. Dunn-Davies and Dr. C. Delan-Forino for help with data analysis, Dr. V. Shchepachev for initial help with sucrose gradients and fractionation, Dr. C. Sayou for RNA isolation and Dr. P. Voigt for access to the BIORAD ChemiDoc system.

## Author contributions

M.-L.W. and D.T. conceived the study. L.P., J.R. and D.T. conceived iRAP. M.-L.W. performed all experiments except iRAP, which was performed and analysed by L.P., with support from T.W.T. for RNA extractions and northern blotting. M.-L.W. and T.W.T. analysed the data. M.-L.W. and D.T. wrote the manuscript, with contributions from all other authors.

## Additional information

**Competing interests:** The authors declare no competing interests.

