## [Peer Review File · Nature Communications]

Reviewers' comments:

Reviewer #1 (Remarks to the Author):

Review for the MS entitled "Molecular interactions between Hel2 and RNA supporting ribosome-associated quality control"

In this paper, the authors developed a novel technique termed identification of RNA associated peptides (iRAP) to identify the RNA binding sites based on in vivo UV crosslinking and mass spectrometry. Using this novel technique, the authors found that Hel2 binds to 18S rRNA and translating mRNAs. The authors applied CRAC (cross-linking and analysis of cDNA) method to tagged Hel2 to precisely identify target sites within the RNA at single-nucleotide resolution. Using these techniques, they found that Hel2 has major binding sites in 18S rRNA and mRNA. Hel2 binds to the small ribosomal subunit in close proximity to known ribosomal protein ubiquitination sites, and deletion of the RNA crosslinking site in Hel2 C-terminal 544 -638 aa region disrupts 18S rRNA association. These are consistent with the previous result to demonstrate the specific ubiquitination of uS10 at K6/8 residues by Hel2. Hel2 also preferentially bound mRNA both upstream and downstream of the termination codon. ASC1 deletion impaired Hel2 18S and mRNA binding. Based on these, they propose that Hel2 is recruited or stabilized on translating 40S ribosomal subunits by interactions with 18S rRNA and Asc1. Overall, this article is of immediate interest in the field and logically presented, and the discussion is fair and accurate. Although several experiments are needed before publication, I recommend the publication of this MS in Nature Communications.

Major Comment

The authors demonstrated that Hel2(1-492) or Hel2(1-543) reduces 18S rRNA association. To evaluate the significance of 18S rRNA association with Hel2 in RQC and NGD, it is necessary to determine the activity of these Hel2 mutants in RQC and NGD. The function of Hel2 in RQC is evaluated by the determination of the levels of the arrest products derived from the reporter genes containing translation-arrest inducing sequence such as R12 or CGA repeats in *ltn1* deletion background. The function of Hel2 in NGD is evaluated by the determination of the levels of the 5' truncated fragments derived from the reporter genes containing translation-arrest inducing sequence such as R12 or CGA repeats in *ski2* deletion background. I also recommend the analysis of alanine substitution mutants of L501/K502 residues in RQC and NGD.

Minor comment

I recommend to confirm the contents of the citation for the function of Hel2 is required for NGD. Saito and co-workers describe the role of K63-linked ubiquitination in translation arrest by CGA repeats. In the paper, it was demonstrated that the full-length Rluc-X-HIS3 mRNA was less markedly affected by deletion of Hel2 and expression of K63R as compared with expression of the full-length protein. Based on these, they proposed that K63 polyubiquitination and Hel2 have more direct roles in protein turnover than mRNA turnover. Moreover, there is no direct detection of the endonucleolytic cleavage product. The authors miss the old publication that demonstrated the function of Asc1 in the degradation of arrest products and an endonucleolytic cleavage by consecutive basic amino acid sequences.

Reviewer #2 (Remarks to the Author):

In this manuscript Winz and colleagues identify a C-terminal RNA-binding domain in Hel2/Rqt RING-type E3 ubiquitin ligase required in no-go decay and ribosome-associated quality control (RQC). Applying crosslinking and analysis of cDNA, the authors identify binding of Hel2 to 18S ribosomal RNA as well as mRNA sequences. Deletion of the Hel2 RNA-binding domain prevented binding to 18S rRNA and reduced Hel2 function.

This is interesting and well conducted work demonstrating Hel2 binding to 18S ribosomal RNA and mRNAs upstream and downstream from the stop codon placing Hel2 on translating and

terminating ribosomes.

There are three issues that should be resolved before considering this manuscript for publication.

One issue is the lack of information on an important method and results presented in this manuscript. The authors should describe iRAP, a method which is mentioned several times as a novel approach leading to the discovery of the C-terminal Hel2 RNA-binding domain. It is unclear how the crosslinked peptides were isolated and identified by mass spectrometry. The authors describe that iRAP identified 1331 unique RNA-interaction sites at single amino acid residue resolution in 324 proteins. Winz and coworkers should provide a detailed description of the iRAP method, results on the reproducibility of this method and the list of the 1331 unique RNA-interaction sites in the 324 RNA-binding proteins that were identified.

Secondly, the authors should examine the functional consequences of deletion of the C-terminus of Hel2 (1-492) on selected transcripts. Does the deletion cause changes in mRNA stability? In particular, the authors should test changes of stability and translation of mRNAs that showed relatively increased binding to Hel2 in the mutant strains.

The third issue refers to recent work by the Tuschl/Sonenberg groups on ZNF598, the human homolog of Hel2 published in Nature Communications. Using PAR-CLIP Garcia et al. showed that the Zinc Finger Protein ZNF598 cross-links to tRNAs, mRNAs and rRNAs, thereby placing the protein on translating ribosomes. The discussion lacks any thorough comparison of the RNA binding patterns of the human and yeast protein and the implication. Furthermore, Winz and coworkers show in Figure 1c a sequence alignment of crosslink-proximal amino acids of Hel2 of different organisms focus on the region that crosslinks to RNA. The human sequence covers the region of amino acids 911 to 935. Interestingly, deleting this region in the human protein does not change its binding activity (Garcia et al. Nature Communications 2017). How do the authors explain this result?

Reviewer #3 (Remarks to the Author):

To respond to stalled protein synthesis, eukaryotes have evolved the Ribosome-associated Quality Control pathway (RQC). RQC is a response that recognizes stalled ribosomes and commits their associated nascent chain to degradation. Hel2 is an E3 ubiquitin ligase that participates in this recognition step by ubiquitylating the 40S ribosomal subunit, a signal that initiates RQC. Despite efforts to characterize the function of Hel2, attempts to purify Hel2-bound ribosomes to establish a structural basis for Hel2 function have not produced a clear density for Hel2. Furthermore, a recent report shows that the human Hel2 ortholog ZNF598 binds to mRNA as well. Thus, how Hel2 associates with the stalled ribosome to ubiquitylate it while still binding mRNA is an open question.

Winz and colleagues ask how Hel2 associates with stalled ribosomes. To resolve this question, the authors turn to a new technique called iRAP that uses crosslinking and mass spectrometry to identify RNA proteins by looking for peptides modified with ribonucleotides. Hel2 was found to bind RNA in these experiments. Using CRAC (a crosslinking and sequencing technique) and other sequencing techniques, the authors determined RNA sequences that were bound by Hel2.

The authors' conclusions were as follows:

Hel2 has a C-terminal RNA binding region that interacts with 18S rRNA to stabilize its association with the 40S.

The position of Hel2 binding explains how it can interact with both 40S and mRNA and how it can ubiquitylate its substrate Rps proteins.

The association of Hel2 with this region of rRNA is scaffolded by Asc1, a 40S ribosomal protein known to be required for RQC.

Hel2 binds to ribosomes and mRNAs and its binding scales with translation level. Hel2 accumulates in the vicinity of the stop codon. This suggests that substrates for the RQC may predominantly be ribosomes stalled at stop codons.

These findings are novel and inform findings from previous work. Identification of the Hel2

binding site, Hel2's binding specificity for mRNAs, and the role of Asc1 resolve important questions. Additionally, demonstrating the utility of iRAP and CRAC techniques is another valuable contribution of this work.

I recommend this paper for publication in Nature Communications after the following changes are made to strengthen the conclusions and some of the presentation:

Throughout the paper, an equivalency is drawn between Hel2 crosslinking to RNAs and Hel2 binding to these RNAs. Crosslinking is a chemical reaction that is influenced by a variety of conditions. Though Hel2 to crosslinking to an RNA is suggestive of an interaction, the main claims the authors make need to be validated with biochemical evidence. At the very least, the authors need to demonstrate that truncation of the putative RNA binding region (Hel2 1-492, Hel2 1-543) and ablation of Asc1 reduces Hel2 association with ribosomes (via sucrose gradient and immunoblot) relative to an unperturbed condition (wt).

The normalization of the data in Figure 7b is overly complex and distorts the data. At this point, the authors have already shown that perturbations that reduce Hel2 crosslinking to 18S rRNA (RNA binding domain truncation or asc1 Δ) reduce the amount of mRNA crosslinking to Hel2 compared to wt (Figure S10). The authors construct a metagene of Hel2 mRNA binding specificity in these conditions, but then normalize the data to the wt case. Because there are so many more reads in the wt case, of course this should shrink the peaks of the other conditions. This is not evidence of a change in binding specificity; to illustrate this point, the authors need to normalize to the each condition (i.e. the area under the curve should be the same for all conditions).

In some cases the authors make qualitative judgements about CRAC data (page 12, second sentence) that are not obvious by looking at the data. In Figure 4A, the Hel2 1-543 and Hel2 1-492 mutants look almost identical. How does one mutant have a loss in crosslinking that is "much more substantial" than the other? How is this consistent with the difference in drug sensitivity mentioned at the bottom of page 12? It would be more clear if the authors either displayed the data in a way that showed the qualitative difference they speak of or if the authors changed the language to reflect the small differences between samples that may actually be observed.

Additionally, the authors should address these specific comments:

Bottom of page 2, sentence beginning with "A further surveillance...": RQC has never been shown to be initiated by misfolding of the nascent chain.

Middle of page 4, bolded section title: Change "C-terminal of Hel2" to "C-terminus of Hel2"

Top of page 7, sentence beginning with "It was, however, notable that the mild...": AZC does not trigger NGD/RQC

Top of page 9, throughout Figure 3 legend: Define what is meant by "hits," as it is important to understanding what this data means. Is "hits" just mapped reads?

Bottom of page 17, sentence beginning with "In mRNA metagene analyses...": State units for "+200 in the wild-type."

Throughout discussion, cite relevant figures.

We thank the referees for their careful reviews of the MS. We have responded to all of the comments and feel that this has significantly improved the paper. Detailed responses are given below.

Reviewer #1 (Remarks to the Author):

Major Comment

The authors demonstrated that Hel2(1-492) or Hel2(1-543) reduces 18S rRNA association. To evaluate the significance of 18S rRNA association with Hel2 in RQC and NGD, it is necessary to determine the activity of these Hel2 mutants in RQC and NGD. The function of Hel2 in RQC is evaluated by the determination of the levels of the arrest products derived from the reporter genes containing translation-arrest inducing sequence such as R12 or CGA repeats in ltn1 deletion background. The function of Hel2 in NGD is evaluated by the determination of the levels of the 5' truncated fragments derived from the reporter genes containing translation-arrest inducing sequence such as R12 or CGA repeats in ski2 deletion background.

I also recommend the analysis of alanine substitution mutants of L501/K502 residues in RQC and NGD.

We have performed all the recommended experiments and results are shown in Figure 7 and Supplementary Figures S10 and S11. These results have strengthened the conclusions of the MS. As far as we are aware, this is the first demonstration of direct involvement of Hel2 in the endonucleolytic cleavage and removal of cleavage fragments in the NGD pathway.

Minor comment

The authors miss the old publication that demonstrated the function of Asc1 in the degradation of arrest products and an endonucleolytic cleavage by consecutive basic amino acid sequences.

We thank the referee for this useful comment. We have not only re-considered the Saito citation and added the missing citation (Kuroha, 2010), but have also re-analysed the published data on Hel2. Hel2 had been cited as an “NGD factor” with a predicted role in NGD, but as noted above, this was apparently not previously directly demonstrated. We have changed the Introduction, Results and Discussion

accordingly.

Reviewer #2 (Remarks to the Author):

There are three issues that should be resolved before considering this manuscript for publication.

One issue is the lack of information on an important method and results presented in this manuscript. The authors should describe iRAP, a method which is mentioned several times as a novel approach leading to the discovery of the C-terminal Hel2 RNA-binding domain. It is unclear how the crosslinked peptides were isolated and identified by mass spectrometry. The authors describe that iRAP identified 1331 unique RNA-interaction sites at single amino acid residue resolution in 324 proteins. Winz and coworkers should provide a detailed description of the iRAP method, results on the reproducibility of this method and the list of the 1331 unique RNA-interaction sites in the 324 RNA-binding proteins that were identified.

**This is an important point. We feel that presentation of the detailed iRAP protocols and the complete set of results are beyond the scope of the of the current paper, which is mainly focused on Hel2. A paper with the requested data has now been published as a pre-print uploaded to bioRxiv and is referenced in the text:
Peil, L., Waghmare, S., Fischer, L., Spitzer, M., Tollervey, D. and Rappsilber, J. (2018) Identification of RNA-associated peptides, iRAP, defines precise sites of protein-RNA interaction. bioRxiv, <http://biorxiv.org/cgi/content/short/456111v1>**

Secondly, the authors should examine the functional consequences of deletion of the C-terminus of Hel2 (1-492) on selected transcripts. Does the deletion cause changes in mRNA stability? In particular, the authors should test changes of stability and translation of mRNAs that showed relatively increased binding to Hel2 in the mutant strains.

We have now conducted additional experiments that address several consequences of C-terminal deletion. The functional consequences on RQC & NGD were addressed using a reporter construct. The biochemical effects on ribosomal association were addressed using sucrose gradients, with detection of tagged, wild-type Hel2 and mutant constructs by Western blot (see referees 1 and 3).

To more directly address the question by this referee, we have also isolated and reverse-transcribed RNA from pooled sucrose fractions gained from wt and truncated Hel2-HTP strains and performed qPCR analyses to determine whether the distribution

of mRNAs that showed relatively increased binding to Hel2 in the mutant strains, to free, monosomal and polysomal fractions changes in the truncation mutant, relative to wt, as a proxy for translation (Supplementary Figure S16c). This analysis did not show any significant differences. Due to time restrictions, we have not been able to also address stability of said transcripts. However, the fact that the steady state levels of mRNAs were unchanged (Supplementary Figure S16b) does not suggest significant changes in stability either, so that we do not expect relatively increased Hel2 recruitment to those transcripts to lead to enhanced no-go decay.

The third issue is refers to recent work by the Tuschl/Sonenberg groups on ZNF598, the human homolog of Hel2 published in Nature Communications. Using PAR-CLIP Garzia et al. showed that the Zinc Finger Protein ZNF598 cross-links to tRNAs, mRNAs and rRNAs, thereby placing the protein on translating ribosomes. The discussion lacks any thorough comparison of the RNA binding patterns of the human and yeast protein and the implication.

A more detailed comparison between our data and data from the the Tuschl and Sonenberg groups has been added to the discussion.

Furthermore, Winz and coworkers show in Figure 1c a sequence alignment of crosslink-proximal amino acids of Hel2 of different organisms focus on the region that crosslinks to RNA. The human sequence covers the region of amino acids 911 to 935. Interestingly, deleting this region in the human protein does not change its binding activity (Garcia et al. Nature Communications 2017). How do the authors explain this result?

The sequence alignment initially presented was generated using only the C-termini of Hel2/ZNF598 and using default settings on Clustal Omega, which aligned the very C-terminus of ZNF598 with an internal region of Hel2. Redoing the alignment with the full-length proteins has given different results, which we think more accurately aligns the sequences. This is shown in the revised version of Figure 1 and of Supplementary Figure S2. In the current alignment, amino acids 567 and 568 of ZNF598 are aligned to the crosslinked amino acids of Hel2. According to Garzia et al., C-terminal truncation to position 749 did not alter RNA crosslinking, while truncation to position 290 did. A similar role for the aligned region in RNA binding could thus be envisioned for ZNF598. However, due to the limited conservation of the Hel2/ZNF598 C-termini, and based on differences in target discrimination (Rps20 being the main target for Hel2 and RPS10 the one for ZNF598) this does not necessarily need to be the case.

Reviewer #3 (Remarks to the Author):

Throughout the paper, an equivalency is drawn between Hel2 crosslinking to RNAs and Hel2 binding to these RNAs. Crosslinking is a chemical reaction that is influenced by a variety of conditions. Though Hel2 to crosslinking to an RNA is suggestive of an interaction, the main claims the authors make need to be validated with biochemical evidence. At the very least, the authors need to demonstrate that truncation of the putative RNA binding region (Hel2 1-492, Hel2 1-543) and ablation of Asc1 reduces Hel2 association with ribosomes (via sucrose gradient and immunoblot) relative to an unperturbed condition (wt).

We have performed the requested experiments and added the results into Figure 6 and Supplementary Figure S9, Results and Discussion.

The normalization of the data in Figure 7b is overly complex and distorts the data. At this point, the authors have already shown that perturbations that reduce Hel2 crosslinking to 18S rRNA (RNA binding domain truncation or *asc1Δ*) reduce the amount of mRNA crosslinking to Hel2 compared to wt (Figure S10). The authors construct a metagene of Hel2 mRNA binding specificity in these conditions, but then normalize the data to the wt case. Because there are so many more reads in the wt case, of course this should shrink the peaks of the other conditions. This is not evidence of a change in binding specificity; to illustrate this point, the authors need to normalize to the each condition (i.e. the area under the curve should be the same for all conditions).

We respectfully disagree. In the case of general loss of RNA binding without a change in targeting, normalization to wt condition should shrink peaks, but should not alter the general binding pattern, which it does (there is only a small, or even no difference right after the Start codon, but the difference between conditions increases over the gene body and towards the Stop codon). In our opinion, a normalization to wt helps here, to show that it is not binding close to the Start codon that is increased, but rather binding closer to the 3'-end that is decreased in the mutant cases. However, to underline our claim, that targeting in general is changed, we agree that normalizing to the conditions can be helpful. We have done this and the resulting graphs are shown in Supplementary Figure S15.

In some cases the authors make qualitative judgements about CRAC data (page 12, second sentence) that are not obvious by looking at the data. In Figure 4A, the Hel2 1-543 and Hel2 1-492 mutants look almost identical. How does one mutant have a loss in crosslinking that is “much more substantial” than the other? How is this consistent with the difference in drug sensitivity mentioned at the bottom of page 12? It would be more clear if the authors either displayed the data in a way that showed the qualitative difference they speak of or if the authors changed the language to reflect the small differences between samples that may actually be observed.

We thank the referee for pointing this out. Regrettably, we misreferred the figure panels, first referring to Figure 4A, then later to 4C, which is the relevant panel. We have corrected this issue and hope that it is now clear which panel shows the loss in crosslinking (deletions).

Additionally, the authors should address these specific comments:

Bottom of page 2, sentence beginning with “A further surveillance...”: RQC has never been shown to be initiated by misfolding of the nascent chain.

We have altered the text.

Middle of page 4, bolded section title: Change “C-terminal of Hel2” to “C-terminus of Hel2”

We have changed this.

Top of page 7, sentence beginning with ‘It was, however, notable that the mild...’: AZC does not trigger NGD/RQC

We have changed this.

Top of page 9, throughout Figure 3 legend: Define what is meant by “hits,” as it is important to understanding what this data means. Is “hits” just mapped reads?

This is correct. We have changed the wording throughout the manuscript.

Bottom of page 17, sentence beginning with “In mRNA metagene analyses...”: State units for “+200 in the wild-type.”

We have added “nt”.

Throughout discussion, cite relevant figures.

We have done this in the revised MS.

REVIEWERS' COMMENTS:

Reviewer #1 (Remarks to the Author):

The authors have addressed most of my previous concerns. I support the publication of manuscript (NCOMMS-18-14881A) entitled "Molecular interactions between Hel2 and RNA supporting ribosome-associated quality control" by Dr. Tollervey and colleagues" in Nature Communications.

Reviewer #2 (Remarks to the Author):

The authors have satisfactorily addressed all comments. There are no issues remaining.

Reviewer #3 (Remarks to the Author):

The experiments performed by the authors have satisfactorily addressed my concerns. The revision enhances the conclusions made by the authors and I therefore recommend publication.